# Multiple introductions of multidrug-resistant typhoid associated with acute infection and asymptomatic carriage, Kenya

Samuel Kariuki[1,2]*[†], Zoe A Dyson[2,3,4,5]†, Cecilia Mbae[1], Ronald Ngetich[1], Susan M Kavai[1], Celestine Wairimu[1], Stephen Anyona[1], Naomi Gitau[1], Robert Sanaya Onsare[1], Beatrice Ongandi[1], Sebastian Duchene[6], Mohamed Ali[7], John David Clemens[8], Kathryn E Holt[4,5], Gordon Dougan[3]

[1]Centre for Microbiology Research, Kenya Medical Research Institute, Nairobi, Kenya; [2]Wellcome Sanger Institute, Wellcome Genome Campus, Cambridge, United Kingdom; [3]Cambridge Institute of Therapeutic Immunology & Infectious Disease (CITIID), Department of Medicine, University of Cambridge, Cambridge, United Kingdom; [4]London School of Hygiene & Tropical Medicine, London, United Kingdom; [5]Department of Infectious Diseases, Central Clinical School, Monash University, Melbourne, Australia; [6]Department of Microbiology and Immunology, The University of Melbourne at The Peter Doherty Institute for Infection and Immunity, Melbourne, Australia; [7]Department of International Health, John's Hopkins University, Baltimore, United States; [8]International Diarrheal Diseases Research Centre, Dhaka, Bangladesh

*For correspondence: samkariuki2@gmail.com

†These authors contributed equally to this work

Competing interest: The authors declare that no competing interests exist.

## Abstract

**Background:** Understanding the dynamics of infection and carriage of typhoid in endemic settings is critical to finding solutions to prevention and control.

**Methods:** In a 3 -year case-control study, we investigated typhoid among children aged <16 years (4670 febrile cases and 8549 age matched controls) living in an informal settlement, Nairobi, Kenya.

**Results:** 148 *S.* Typhi isolates from cases and 95 from controls (stool culture) were identified; a carriage frequency of 1 %. Whole-genome sequencing showed 97 % of cases and 88 % of controls were genotype 4.3.1 (Haplotype 58), with the majority of each (76% and 88%) being multidrug-resistant strains in three sublineages of the H58 genotype (East Africa 1 (EA1), EA2, and EA3), with sequences from cases and carriers intermingled.

**Conclusions:** The high rate of multidrug-resistant H58 *S.* Typhi, and the close phylogenetic relationships between cases and controls, provides evidence for the role of carriers as a reservoir for the community spread of typhoid in this setting.

**Funding:** National Institutes of Health (R01AI099525); Wellcome Trust (106158/Z/14/Z); European Commission (TyphiNET No 845681); National Institute for Health Research (NIHR); Bill and Melinda Gates Foundation (OPP1175797).

## Introduction

Typhoid fever, caused by *Salmonella enterica* serovar Typhi (*S.* Typhi) is estimated to involve ~21.7 million illnesses and 216,000 deaths annually *Crump and Mintz, 2010*; *Mogasale et al., 2014*, with most of these occurring in lower and middle-income countries. In Africa, overall typhoid is now estimated to

have an average annual pooled incidence rate of 112.1 (95% CI, 46.7–203.5) cases per 100,000 people *Hopewell and Graham, 2014*; *Marchello et al., 2019* with a case fatality rate (CFR) of 5.4 % (2.7–8.9) *Marchello et al., 2020*.

Control of typhoid is impeded by asymptomatic carriage, which historically was estimated to account for 2–5% of individuals infected *Levine et al., 1982*; *Parry et al., 2002*; *Thanh Duy et al., 2020*. However, there is a paucity of recent data on the frequency of carriers in different settings including sub-Saharan Africa (SSA) as well as the extent to which they contribute to disease transmission *Gauld et al., 2018*. A recent modelling study using data generated in Blantyre, Malawi, identified multidrug resistant (MDR) *S. Typhi* and/or the emergence of the lineage known as H58 (genotype 4.3.1) as a primary driver of an increasing number of typhoid fever cases. In this study, an estimated 45–95% of typhoid transmission was attributed to carriers *Pitzer et al., 2015*; *Saad et al., 2017*. *S.* Typhi H58 *Wong et al., 2015* is a globally disseminated clade frequently associated with MDR (defined as resistance to chloramphenicol, ampicillin and co-trimoxazole) and an increasing frequency of reduced susceptibility to fluoroquinolones. H58 *S.* Typhi are rapidly displacing other lineages in many endemic areas *Wong et al., 2015*; *Feasey et al., 2015*; *Kariuki et al., 2010*; *Park et al., 2018* and a new subclade that is extensively drug resistant (XDR), displaying resistance to ciprofloxacin and fluoroquinolones in addition to MDR, has been described in Pakistan *Klemm et al., 2018*.

Recent reports of epidemics of typhoid fever in SSA suggest that the disease may be becoming more widespread in the region *Crump and Mintz, 2010*; *Park et al., 2018*; *Hendriksen et al., 2015*; *Lutterloh et al., 2012*; *Marks et al., 2017*; *Neil et al., 2012*. In Kenya, the rapid growth of population has led to a huge rural-to-urban migration with people increasingly living in informal settlements where clean water and good sanitation are a major challenge *Kyobutungi et al., 2008*; *Mberu et al., 2016*. The incidence of typhoid in one such informal settlement, Kibera in Nairobi, was estimated at 247 cases per 100,000 with the highest rates in children 5–9 years old (596 per 100,000) *Breiman et al., 2012*. For the last two decades, the majority of cases of typhoid in Kenya have been MDR, with reduced susceptibility to fluoroquinolones rising in frequency *Kariuki et al., 2010*; *Park et al., 2018*; *Mutai et al., 2018*. Previously, we showed that *S. Typhi* H58 gained a foothold in Kenya in the 1990s, constituting >75% of the circulating *S. Typhi* we have characterised since 2001 *Kariuki et al., 2010*. Two H58 lineages were detected; lineage I being isolated between 1988 and 2008 and lineage II from 2004 onwards. We have previously observed carriage rates of 6 % in households where typhoid cases were detected *Kariuki et al., 2010*, however these *S. Typhi* isolates were not characterised genetically and the role of asymptomatic carriers in transmission dynamics of typhoid in the community is still poorly understood. Over the past 7 years, we have been intensively studying typhoid and other invasive bacterial diseases in Mukuru, an informal settlement 15 km east of the city of Nairobi, Kenya. The prevalence of *S.* Typhi infections among 16,236 children was 1.4 % (CI: 1.2–1.6%), and higher amongst males (1.8% vs 1.2 % for females), with a high proportion of infections noted among older children 5–8 years in age *Mbae et al., 2020*. Risk factors predictive of *S.* Typhi infection in Mukuru were multiple but were predominantly associated with contaminated water sources and sanitation issues *Mbae et al., 2020*. Here, we analysed typhoid cases in Mukuru clinically and microbiologically, and identified frequent asymptomatic carriage among children below 16 years of age. By exploiting whole genome sequencing (WGS) and geospatial mapping we characterised the population structure and transmission dynamics of *S.* Typhi in this location.

## Materials and methods

**Key resources table**

| Reagent type (species) or resource | Designation | Source or reference | Identifiers | Additional information |
|---|---|---|---|---|
| gene (*Salmonella enterica* serotype Typhi – *S. Typhi*) | Wild type | This study | PRJEB19289 | ENA sequence accession bank |
| Strain, strain background (*S. Typhi*) Wild type | Wild type | This study | *Supplementary file 1* | |

*Continued on next page*

*Continued*

| Reagent type (species) or resource | Designation | Source or reference | Identifiers | Additional information |
|---|---|---|---|---|
| Strain, strain background (*Escherichia coli*) | ATCC | ATCC25922 | https://www.atcc.org/ › products › 25,922 | |
| Sequence-based reagent | Primer vi-F | This paper | PCR primers | GTTATTCAGC ATAAGGAG |
| Sequence-based reagent | Primer Vi-R | This paper | PCR Primers | CTTCCATACCA CTTTCCG |
| Sequence-based reagent | Primer prt-F | This paper | PCR Primers | CTTGCTATGGA AGACATAACGAACC |
| Sequence-based reagent | Primer prt-R | This paper | PCR Primers | CGTCTCCATCA AAAGCTCCATAGA |
| Commercial assay or kit | Bactec Media | Becton-Dickinson | BACTEC 9050 Blood Culture System | Blood culture media |
| Commercial assay or kit | Selenite F/ MacConkey | Oxoid Ltd http://www.oxoid.com | CM0395/ CM0007 | Selective enrichment/ selective media |
| Commercial assay or kit | *Salmonella-Shigella* agar | Oxoid Ltd http://www.oxoid.com | CM0099 | Selective agar |
| Commercial assay or kit | Salmonella antisera | Murex Diagnostics, Dartford, UK https://www.dnb.com | | Salmonella typing antisera |
| Commercial assay or kit | Wizard Genomic DNA Extraction Kit | https://worldwide. promega.com | Whole Genome DNA extraction | Cat#A1120 |
| Chemical compound, drug | Antimicrobial susceptibility test discs in cartridges | Oxoid Ltd http://www.oxoid.com | Assorted antimicrobial discs for susceptibility testing | |
| Software, algorithm | Kraken | *Genome Biol* 2014; 15: R46–12. | ultrafast metagenomic sequence classification using exact alignment | |
| Software, algorithm | Multi-locus sequence typing (MLST) | *Genome Med* 2014; 6: 90. | Rapid genomic surveillance for public health and hospital microbiology labs. | |
| Software, algorithm | BIGSdb software | *BMC Bioinformatics* 2010; 11: 595. | Scalable analysis of bacterial genome variation at the population level. | Genomic analysis software |
| Software, algorithm | Pathogen-watch for AMR prediction. | *Nat Commun* 2021; 12: 2879–12. | AMR prediction software analysis | |
| Software, algorithm | Maximum Likelihood Analytical Tool | *Bioinformatics* 2014; 30: 1312–3. | RAxML (v8.2.9) version 8 | A tool for phylogenetic analysis and post-analysis of large phylogenies |
| Software, algorithm | MicroReact Tool | (https://microreact.org/ project/wViqma RdZuFVEb6yk4i1jU) | Interactive global H58 phylogeny | |
| Software, algorithm | Bandage assay | *Bioinformatics* 2015; 31: 3350–2. | Interactive visualisation of de novo genome assemblies. | |
| Software, algorithm | ISMapper | *BMC Genomics* 2015; 16: 667. | Genomics tool for phylogenetics | |

## Study site

Mukuru informal settlement is situated East of Nairobi city, about 15 km from the city centre. It is one of the largest slums in the city with a population of around 250,000 people *KNBS KNBOS, 2009*. The informal settlement is made up of improvised temporary dwellings often made from scrap materials,

such as corrugated metal sheets, plywood, and polythene-sheets *Olack et al., 2011*. In addition to poverty, a number of factors associated with informal settlements, including overcrowding, substandard housing, unclean and insufficient quantities of water, and inadequate sanitation, contribute to a high incidence of infectious diseases and increased mortality among children under five years *Kyobutungi et al., 2008*; *Mutisya and Yarime, 2011*. Mukuru informal settlement is divided into eight villages; Mukuru Lunga-Lunga, Mukuru kwa Sinai, Mukuru kwa Ruben, Mukuru kwa Njenga, Mukuru Kayaba, Fuata Nyayo, Jamaica, and Mukuru North. This study was carried out in two of the large villages, Mukuru kwa Njenga and Mukuru kwa Ruben, with a combined population of 150,000. Spatial mapping of the two villages was conducted using the Universal Transverse Mercator system *Tinline and Gregory, 1988*, and patient details collected as described previously *Mbae et al., 2020*.

The two villages in the informal settlement are served by three outpatient clinics: Ruben Health Centre located in the Ruben village (zone named Simba cool, serves approximately 30 % of the population), Missionaries of Mary Located in Kwa Njenga village (zone named Vietnam, serves approximately 45 % of the population), and County Government Clinic in Kwa Njenga village (Zone named MCC and serves approximately 25 % of the population). The fourth site, Mbagathi District Hospital, is located on the western side of Nairobi city, 5 km from city centre and was used as a referral facility. Participants living outside of the mapped demographic surveillance site (DSS) who came to seek medical services in any of the three study site health facilities or at Mbagathi District Hospital were included for the purpose of tracking typhoid cases and carriers treated at the facilities, but are reported separately in the Results section.

## Recruitment of clinical typhoid fever cases and asymptomatic typhoid carriers

Typhoid fever cases and asymptomatic carriers presented in this study were identified and recruited as part of a larger study on surveillance and genomics of invasive *Salmonella* disease in children and young adults less than 16 years of age *Mbae et al., 2020*; *Kariuki et al., 2020*. Children presenting as outpatients at the three study clinics and Mbagathi District Hospital between August 2013 and November 2016 were triaged to identify those with fever, headache and/or diarrhoea for recruitment into the study as potential cases. Patients with current fever ( ≥ 38 °C) and reportedly febrile for ≥3 days were considered potential typhoid cases and assessed via blood or stool culture. The primary typhoid case definition (data presented in *Table 1*) was children aged 0–16 years with ≥3 days fever ≥38 °C and positive blood or stool culture for *S.* Typhi (see bacterial culture methods below).

During the study period, age-matched controls were recruited from children without current fever or diarrhoea attending the same health facilities for healthy mother and child clinics (e.g. for vaccination and nutritional advice). Those with *S.* Typhi-positive stool culture were designated as asymptomatic typhoid carriers as described previously *Mbae et al., 2020*. Hence, the inclusion criteria for asymptomatic typhoid carriers (data presented in *Table 1*) were children aged 0–16 years with no diarrhoea, no current fever, and no recent fever history, with stool culture positive for *S.* Typhi (see bacterial culture methods below). The total number of participants was computed on the basis of a 4 % prevalence rate of typhoid from previous study *Kariuki et al., 2010*. (A structured questionnaire was used to collect demographic data for both cases and controls recruited into the study as described previously *Mbae et al., 2020*).

All isolates cultured from participants and identified as *Salmonella* were archived and later revived for WGS as detailed below. The sequence data revealed some mis-identification of *Salmonella* serotypes (*Figure 1*, *Supplementary file 1*), hence for genomic analyses we included all cases and controls whose cultures were found to be *S.* Typhi positive by WGS rather than those identified as *S.* Typhi positive by serotype in the microbiology laboratory.

## Bacterial culture

For blood culture, 1–3 mL for children < 5 years of age and 5–10 mL for those 5–16 years of age was collected in a syringe, placed into Bactec media bottles (Becton-Dickinson, New Jersey, USA), incubated at 37 °C in a computerised BACTEC 9,050 Blood Culture System (Becton-Dickinson), and subcultured after 24–48 hr onto blood, chocolate and MacConkey agar (Oxoid, Basingstoke, UK) plates. All isolates, whether from cases or carriers were cultured on selenite F (Oxoid) broth aerobically at 37 °C overnight. Broth cultures were then subcultured on MacConkey agar and *Salmonella-Shigella*

**Table 1.** Culture-positive typhoid cases and asymptomatic carriers.

Note the values reported for logistic regressions are from multivariate models including all indicated covariates, fit separately for cases and controls.

| Participants tested, N | **4,670** | **8,549** |
|---|---|---|
| Male, N (%) | 2,497 (53.5%) | 4,260 (49.8%) |
| Female, N (%) | 2,173 (46.5%) | 4,289 (50.2%) |
| *S. Typhi* culture positive, N (%) | 148 (3.2%) | 95 (1.1%) |
| Male, N (%) | 99 (4.0%) | 49 (1.15%) |
| Female, N (%) | 49 (2.3%) | 46 (1.1%) |
| WGS confirmed *S. Typhi*, N (%) | 100 (2.1%) | 55 (0.64%) |
| Logistic regression for *S. Typhi* culture positive | | |
| Year of isolation, OR (p-value) | 1.19 (0.072) | 0.94 (0.586) |
| Male Sex, OR (p-value) | 1.81 (0.0008*) | 1.08 (0.699) |
| Age in years, OR (p-value) | 1.08 (0.0005*) | 1.02 (0.403) |
| Logistic regression for *S. Typhi* culture positive, males only | | |
| Year of isolation, OR (p-value) | 1.19 (0.147) | 1.09 (0.576) |
| Age in years, OR (p-value) | 1.11 (0.0001*) | 1.06 (0.082) |
| Logistic regression for *S. Typhi* culture positive, females only | | |
| Year of isolation, OR (p-value) | 1.19 (0.296) | 0.81 (0.158) |
| Age in years, OR (p-value) | 1.03 (0.551) | 0.98 (0.534) |
| Logistic regression for *S. Typhi* WGS positive | | |
| Year of isolation, OR (p-value) | 1.15 (0.209) | 0.59 (0.0003*) |
| Male Sex, OR (p-value) | 1.59 (0.028*) | 0.93 (0.780) |
| Age in years, OR (p-value) | 1.11 (0.0001*) | 1.03 (0.326) |
| Logistic regression for *S. Typhi* WGS positive, males only | | |
| Year of isolation, OR (p-value) | 1.18 (0.261) | 0.74 (0.133) |
| Age in years, OR (p-value) | 1.15 (0.00002*) | 1.07 (0.131) |
| Logistic regression for *S. Typhi* WGS positive, females only | | |
| Year of isolation, OR (p-value) | 1.12 (0.558) | 0.48 (0.0003*) |
| Age in years, OR (p-value) | 1.03 (0.518) | 1.0 (0.92) |

agar (Oxoid) and incubated at 37 °C overnight. Blood and stool isolates were identified using a series of standard biochemical and serological tests as described previously *Mbae et al., 2020*. Briefly, colonies of *S. Typhi* identified from biochemical and PCR testing were subjected to serological identification as *S. Typhi* using the slide agglutination technique and applying monovalent antisera (Murex Diagnostics, Dartford, UK). A drop of the antisera to be tested was mixed with a bacterial smear on a slide to observe for the presence or absence of agglutination in two minutes.

## Antimicrobial susceptibility testing

Antimicrobial susceptibility testing was performed using the disk diffusion technique for ampicillin 10 µg, tetracycline 30 µg, co-trimoxazole 25 µg, chloramphenicol 30 µg, cefpodoxime 30 µg, ceftazidime 30 µg, ceftriaxone 30 µg, cefotaxime 30 µg, ciprofloxacin 5 µg, and nalidixic acid 10 µg as described previously *Mbae et al., 2020*. Results were interpreted according to the 2017 guidelines provided by the Clinical and Laboratory Standards Institute (CLSI) *Patel et al., 2015*.

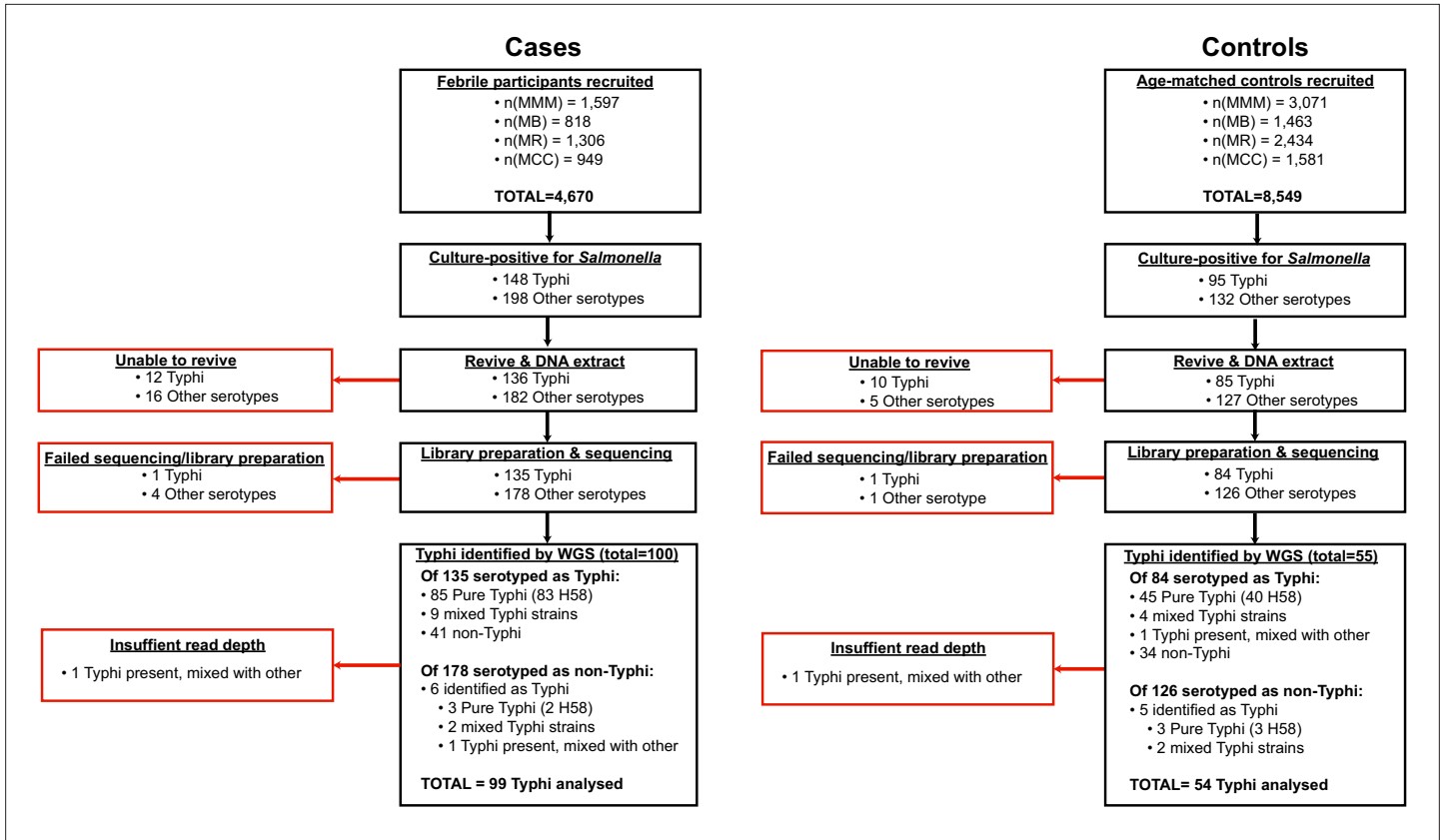

**Figure 1.** Flow chart of samples collected and analysed. Red boxes indicate bacterial isolates that could not be included in downstream genetic analyses, grouped by reason for exclusion.

## Whole genome sequencing

All *Salmonella* isolated from cases and controls were subcultured at the end of the study for DNA extraction and WGS. These included 243 cultures identified as *S*. Typhi from cases (85 from blood and 63 from stool) and 95 from controls (all from stool) (see *Figure 1*, (1) *Supplementary file 1*), which are the subject of this study (non-typhoidal *Salmonella* data is reported elsewhere *Kariuki et al., 2020*). Twelve *S*. Typhi case isolates and 10 control isolates could not be revived and were not further analysed. DNA was extracted using the Wizard Genomic DNA Extraction Kit (Promega, Wisconsin, USA) and shipped on ice to the Wellcome Sanger Institute for sequencing using the Illumina platform as described previously *Park et al., 2018*. A total of 217 *S*. Typhi DNA samples were successfully sequenced (two were of insufficient quality to construct sequencing libraries, or failed sequencing). Non-*S*. Typhi bacterial DNA sequences were detected in 75 samples (34.6 %; organisms detected are shown in *Supplementary file 1*), and 11 sequences originally identified as other *Salmonella* serotypes were later found to be *S*. Typhi with genomic data. Two sequences showed the presence of *S*. Typhi, but at low depth, and were subsequently omitted from further genomic analyses, leaving genome data for 153 *S*. Typhi isolates for further analysis.

The taxonomic identities of non-typhoidal *Salmonella* spp. were determined using Kraken (v0.10.6) *Wood and Salzberg, 2014*, multi-locus sequence typing (MLST) with SRST2 *Inouye et al., 2014*; *Jolley and Maiden, 2010*, as well as the *Salmonella* In Silico Typing Resource (SISTR) and Speciator (both available via Typhi Pathogenwatch; https://pathogen.watch/) *Argimón et al., 2021*.

## Phylogenetic and SNP analysis of *S. Typhi* isolates

For SNP analysis, paired-end reads from 153 *S*. Typhi isolates were mapped to the reference sequence of *S*. Typhi CT18 (accession number: AL513382) *Parkhill et al., 2001* using the RedDog mapping pipeline (v1beta.10.3), available at https://github.com/katholt/reddog (copy archived at swh:1:rev:90707ace56d6189997526273cf97013d84b84d90; *Holt, 2021a*) and detailed in

supplementary methods. Read alignments were used to assign isolates to previously defined lineages according to the extended genotyping framework *Wong et al., 2016*; *Britto et al., 2018* with the GenoTyphi pipeline (available at https://github.com/katholt/genotyphi, copy archived at swh:1:rev:f-489d0e004e30bdca683c76c25ac7f4d79162791; *Holt, 2021b*). Unique SNPs defining three novel lineages were identified from the genome-wide SNP allele table and added to the GenoTyphi scheme to facilitate easy identification of these lineages in future studies (details in supplementary methods and results).

Phylogenetic analyses were restricted to WGS-confirmed pure cultures of *S.* Typhi H58 (genotype 4.3.1, n = 128). For some analyses, an additional 1076 *S. Typhi* H58 genomes from previously published WGS studies of global and African isolates *Wong et al., 2015*; *Park et al., 2018*; *Wong et al., 2016*; *Pham Thanh et al., 2016* were also included for context, along with 61 non-H58 genomes for phylogenetic outgroup rooting, using the same mapping approach detailed above (see *Supplementary file 2* for full list of genomes analysed and their public data accessions). SNPs called in phage regions or repetitive sequences were filtered from the alignment (details in supplementary methods). Analysis with Gubbins (v2.3.2) identified two regions affected by recombination (coordinates 954,115–970,731 (genes STY0961-STY0976), and 1,438,676–1,467,273 (genes STY1485-STY1508) in the CT18 reference genome), which were excluded from the alignment *Croucher et al., 2015*. This resulted in a final set of 8635 SNPs. From this global alignment we extracted a separate SNP alignment for the set of 239 Kenyan *S. Typhi* 4.3.1 genomes (n = 128 from this study and n = 111 from published studies, see *Supplementary file 3*; *Feasey et al., 2015*; *Yu et al., 2016*), the resulting alignment of length 489 SNPs was used for temporal analyses (described below and in supplementary methods).

Maximum likelihood (ML) phylogenetic trees were inferred from SNP alignments using RAxML (v8.2.9) *Stamatakis, 2014* (as detailed in supplementary methods) and the resulting trees were visualised using Microreact (interactive global H58 phylogeny available at: https://microreact.org/project/wViqmaRdZuFVEb6yk4i1jU) *Argimon et al., 2016*.

Pairwise SNP distances were calculated from the SNP alignment using the dist.dna() function in the R package *ape* (v5.4.1) *Paradis et al., 2004*. Terminal branch lengths were extracted from phylogenies using R package *ggtree* (v2.2.4) *Yu et al., 2016*. Non-synonymous mutations occurring in terminal branches were detected using SNPPar (v0.4.2dev) *Edwards et al., 2020* and grouped by function based on the gene in which they were found, according to the functional classification scheme in the genome annotation of *S.* Typhi CT18 *Thanh Duy et al., 2020*; *Parkhill et al., 2001*.

## Phylodynamic analysis

To investigate temporal signal and date the introduction of *S. Typhi* H58 into Kenya based on the 239 available Kenyan genomes (n = 128 from this study, and n = 111 from previous studies *Park et al., 2018*; *Wong et al., 2016*), we used several methods. First, we used TempEst (v1.5.1) *Rambaut et al., 2016* to assess temporal structure (i.e. clock-like evolution) by conducting a regression analysis of the root-to-tip branch distances of the ML tree as a function of sampling date, and later a date-randomisation test (full details of temporal signal assessment and model selection are provided in supplementary methods). To estimate divergence dates for the three *S. Typhi* H58 sublineages we detected in Kenya (EA1-3), we used BEAST (v1.10) *Suchard et al., 2018* to fit a phylodynamic model to the SNP alignment and isolation dates as described in supplementary methods. The resultant MCC tree was visualised using *ggtree* (v2.2.4) *Yu et al., 2016* and Microreact *Argimon et al., 2016* (interactive phylogeny available at: https://microreact.org/project/I2KUoasUB).

## Genomic determinants of antimicrobial resistance

The read mapping-based allele typer SRST2 (v0.2.0) *Inouye et al., 2014* was used to detect the presence of plasmid replicons (PlasmidFinder database *Carattoli et al., 2014*) and antimicrobial resistance (AMR) genes (ARGannot database *Gupta et al., 2014*). Where AMR genes were observed without evidence of a known AMR plasmid, raw read data was assembled using Unicycler (v0.4.7) *Wick et al., 2017* and then examined using Bandage (v0.8.1) *Wick et al., 2015* to confirm the chromosomal location and composition of AMR-associated transposons. As the Tn*2670*-like composite transposon commonly associated with the acquisition of MDR genes in *S.* Typhi is mediated by IS*1* translocation *Wong et al., 2015*, ISMapper (v2.0) *Hawkey et al., 2015* was also used to identify the location of IS*1* insertion sequences in the *S.* Typhi chromosome as described in supplementary methods. SRST2

was used to determine IncHI1 plasmid MLST (multi-locus sequence types) sequence types (pMLST), and minor alleles were identified by mapping to the plasmid pAKU1 reference sequence (accession number AM412236) in the same manner as described above for the *S.* Typhi chromosome (details in supplementary methods and *Supplementary files 4-5*). Point mutations located within the quinolone resistance determining region (QRDR) of genes *gyrA, gyrB,* and *parC* associated with reduced susceptibility to fluoroquinolones *Pham Thanh et al., 2016* were detected using GenoTyphi *Wong et al., 2016*; *Britto et al., 2018* as detailed in supplementary methods.

## Statistical and spatial analysis

All statistical analyses unless otherwise stated were carried out using R (v4.0.2). Details of specific functions within R packages used for individual analyses are available in supplementary methods.

## Nucleotide sequence and read data accession numbers

Raw Illumina sequence reads have been submitted to the European Nucleotide Archive (ENA) under accession PRJEB19289. Individual sequence accession numbers are listed in *Supplementary file 1*.

## Results

### Detection of *S. Typhi* cases and asymptomatic carriers

From August 2013-November 2016, a total of 4670 febrile children were recruited across the four study sites and subjected to blood and/or stool culture. *S.* Typhi was identified in cultures from 148 children (3.2%); the annual rate was steady over the study period but significantly higher amongst males (4.0% vs 2.3%, p = 0.0008, see *Table 1*). The odds of *S.* Typhi positive culture increased significantly with age (OR 1.08, p = 0.0005) but the effect was restricted to males (see *Table 1*), amongst whom the isolation rate was 1.3 % in those ≤1 year, 2.0 % in those aged 1–7 years, and 3.4 % in those >7 years old (compared with 0.95%, 1.1%, and 0.94%, respectively amongst females). A total of 8549 age-matched control participants (with no current diarrhoea and no recent fever history) were recruited and subjected to stool culture. *S.* Typhi was identified in cultures from n = 95 (1.1%); these are considered asymptomatic carriers. *S.* Typhi culture positivity amongst controls was not significantly

**Table 2.** Genotypes and AMR profiles for 153 sequenced *S. Typhi* isolates.

Percentages indicate genotype frequencies amongst cases or controls (first two columns); or frequency of antimicrobial resistance determinants amongst isolates of a given genotype (remaining columns). MDR, multi-drug resistant; L1, lineage I; L2, lineage II.

| | | | MDR | | GyrA mutation | | | GyrB mutation |
|---|---|---|---|---|---|---|---|---|
| Genotype | Cases | Controls | Plasmid | Chromosome | S83F | S83Y | D87G | S464F |
| All | 99 | 54 | 83 | 33 | 3 | 17 | 2 | 75 |
| 2.2.2 | 0 | 1 (1.9%) | 0 | 0 | 0 | 0 | 0 | 0 |
| 2.5.0 | 1 (1.0%) | 2 (3.7%) | 0 | 0 | 0 | 0 | 0 | 0 |
| 3.0.0 | 2 (2.0%) | 1 (1.9%) | 0 | 0 | 0 | 0 | 0 | 0 |
| 4.1.1 | 0 | 1 (1.9%) | 0 | 0 | 0 | 0 | 0 | 0 |
| 4.3.1 (H58) | 96 (97%) | 49 (91%) | 83 (57%) | 33 (23%) | 4 (2.8%) | 19 (13%) | 2 (1.4%) | 75 (51.7%) |
| *H58 subgroups* | | | | | | | | |
| EA1 (L1) | 35 (35%) | 20 (37%) | 29 (53%) | 17 (31%) | 4 (7.3%) | 2 (3.6%) | 2 (3.6%) | 2 (3.6%) |
| EA2 (L2) | 46 (46%) | 27 (50%) | 54 (74%) | 0 | 0 | 0 | 0 | 73 (100%) |
| EA3 (L2) | 15 (15%) | 2 (3.7%) | 0 | 16 (94%) | 0 | 17 (100%) | 0 | 0 |

associated with age or sex and was stable over the study period (see *Table 1* and *Supplementary file 6*). Significant associations identified from culture data were also observed when using WGS confirmed infections only, and no significant statistical association was found between phenotypic or genotypic AMR patterns and case/control status, age, or sex.

## Global population structure and antimicrobial resistance profiles of kenyan *S. Typhi*

The presence of *S*. Typhi was confirmed by WGS in 94 cases (64%) and 50 controls (53%) that were originally identified as *S*. Typhi via microbiological culture (*Figure 1*, *Supplementary file 1*). *S. Typhi* genotype 4.3.1 (H58) was dominant throughout the study (n = 145, 95%), amongst both cases and controls (*Table 2*). Five other genotypes were detected: 2.2.2 (n = 1), 2.5.0 (n = 3), 3.0.0 (n = 3), and 4.1.1 (n = 1), see *Table 2*.

The few non-H58 isolates (*Table 2*) lacked any known AMR determinants. In contrast, the majority of H58 isolates were MDR (n = 116, 80%), often carrying acquired genes conferring resistance to ampicillin (*bla*TEM-1), chloramphenicol (*catA1*), co-trimoxazole (*dfrA7* plus *sul1* and/or *sul2*), and streptomycin (*strAB*). In 33 genomes (23 % of H58), these genes were carried by a Tn*2670*-like complex transposable element inserted in the chromosome as reported previously in the region *Wong et al., 2015*; *Feasey et al., 2015*; *Park et al., 2018*. The remaining 83 MDR genomes (57 % of H58) carried a closely related Tn*2670*-like transposon located within an IncHI1 plasmid, which in all but one isolate also carried an additional tetracycline resistance gene (*tetB*). The IncHI1 plasmids were genotyped as plasmid sequence type 6 (PST6), which is associated with MDR H58 in East Africa and South Asia *Wong et al., 2015*; *Park et al., 2018*; *Holt et al., 2011*. We compared single-nucleotide variant haplotypes for these plasmids with those from 534 IncHI1 PST6 plasmids sequenced previously from African and global studies (all of which were carried by H58 *S*. Typhi hosts, see *Supplementary file 4*). The wildtype PST6 plasmid haplotype was detected in *S*. Typhi hosts of all 6 H58 genotypes, whereas the derived plasmid haplotypes were each detected in a single *S*. Typhi host genotype (*Figure 2— figure supplement 1*). This is consistent with ongoing microevolution of the PST6 plasmid within *S*. Typhi lineages since the acquisition of the plasmid by the mrca of *S*. Typhi H58, but shows no evidence of transfer of plasmid haplotypes between *S*. Typhi lineages. Thus, the observed AMR phenotypes in our study site (n = 128 H58 and n = 8 non-H58 genome sequences) corresponded to the presence of known molecular determinants of AMR and mobile genetic elements. Estimates of sensitivity and specificity of AMR genotyping are presented in *Supplementary file 7* and supplementary results. No statistical association was observed between the presence of MDR genes or QRDR mutations shown in *Table 2* and case/control status, age, or sex.

## Local subpopulations of *S. Typhi* H58

*S. Typhi* H58 (genotype 4.3.1) can be subdivided into lineages I (genotype 4.3.1.1) and II (genotype 4.3.1.2). Lineage II was more common in this setting than lineage I: n = 90 (62.1% of H58) vs n = 55 (37.9%). Examination of the global phylogeny (*Figure 2*, and online interactive version https://microreact.org/project/wViqmaRdZuFVEb6yk4i1jU) revealed all H58 lineage I isolates from this study shared a most recent common ancestor (mrca) whose descendants form a monophyletic clade that exclusively comprised *S*. Typhi from East African countries (see *Figure 2*,), here defined as H58 sublineage EA1 (East Africa 1) with genotype designation 4.3.1.1.EA1 (labelled in *Figure 2*).

*S*. Typhi H58 lineage II (genotype 4.3.1.2) isolates from our study belonged to two distinct clades of the global phylogeny (*Figure 2*), which were each exclusively populated by East African isolates. The largest of these clades (n = 80 isolates, of which 81.3 % derive from the current study) formed a monophyletic group nested within a deeper clade of diverse South Asian isolates (see *Figure 2*), and corresponds to the previously reported introduction of H58 lineage II into Kenya from South Asia *Wong et al., 2015*. This lineage, here defined as H58 sublineage EA2 (East Africa 2) is designated genotype 4.3.1.2.EA2 (labelled in *Figure 2*). The smaller East African H58 lineage II clade (n = 43 isolates) is designated genotype 4.3.1.2.EA3 (labelled in *Figure 2*) and comprised two sister clades, separated by ≥13 SNPs: one involving isolates from Kenya (n = 13, all from this study) and the other isolates from Uganda (n = 30), which accounted for 100 % of the typhoid burden at the Ugandan site where they were identified (see *Figure 2*). All three East African H58 genotypes have been added to

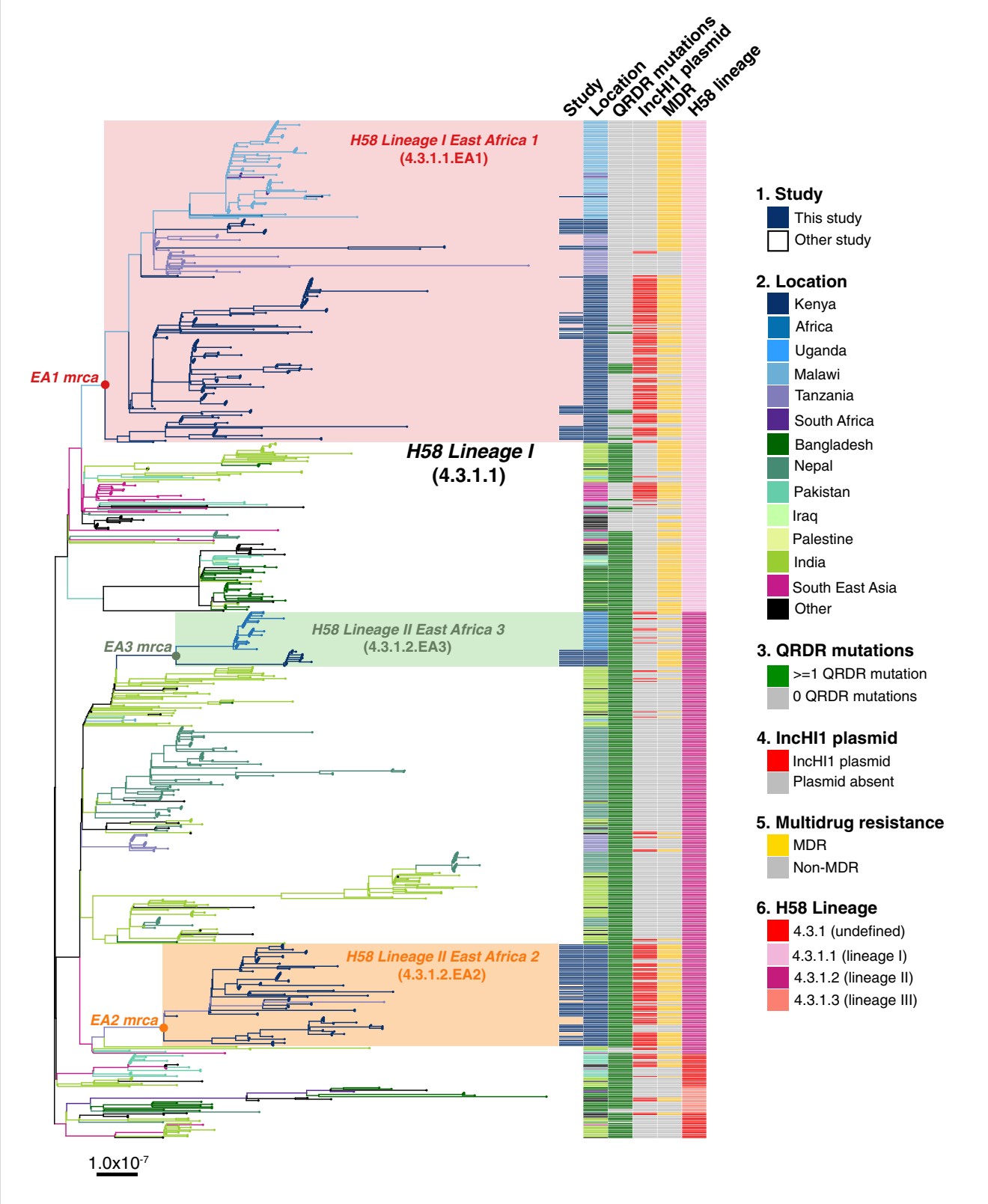

**Figure 2.** Global population structure of H58 (4.3.1) *S. Typhi* showing Kenyan isolates cluster into three East African clades. Whole genome phylogeny of 1,204 H58 isolates, including all available Kenyan genomes (n = 128 from this study, n=111 from prior studies) and globally distributed genomes for context (n=965, see Methods). Branch lengths are in substitutions per core-genome site, branches are coloured to indicate geographical origin (see inset legend), shaded boxes highlight the three East African H58 clades defined in this study. Colour bars to the right indicate (as per inset legend): 1,

*Figure 2 continued on next page*

*Figure 2 continued*

Kenyan strains isolated and sequenced during this study; 2, geographical location; 3, mutation(s) in the quinolone resistance determining region (QRDR) of genes *gyrA, gyrB,* and *parC*; 4, presence of multidrug resistance (MDR) IncHI1 plasmid; 5, presence of MDR genes; 6, H58 lineage. Interactive version available at https://microreactorg/project/wViqmaRdZuFVEb6yk4i1jU.

The online version of this article includes the following figure supplement(s) for figure 2:

**Figure supplement 1.** *S. Typhi* IncHI1 PST6 plasmid minimum spanning tree.

the GenoTyphi scheme using unique marker SNPs and further details on these are provided in supplementary results.

The three East African H58 subgroups circulating in our setting all had high rates of MDR (84%, 74%, and 94%, respectively); however, in EA2, MDR was exclusively associated with the PST6-IncHI1 plasmid, and in EA3 exclusively with the chromosomal insertion (see *Table 2*, *Figure 2*). In EA1, most MDR was associated with the PST6-IncHI1 plasmid. However, a subclade of isolates (associated with spread to Tanzania and Malawi) carried the chromosomal insertion instead (see *Table 2*, *Figure 2*, , supplementary methods).

## Distribution of *S. Typhi* genotypes amongst individuals

No statistically significant differences in genotype distribution were observed between cases and controls (p = 0.077, using Chi-squared test, data in *Table 2*), or between males and females (p = 0.37, using Chi-squared test, data in *Supplementary file 8*), consistent with symptomatic and asymptomatic infections being drawn from the same general circulating pool of pathogens. The distribution of genotypes amongst cases varied by age group (p = 0.01, using Chi-square test), with the frequency of EA1 declining with age and the overall diversity increasing with age (*Table 3*). No significant differences in age groups was evident amongst controls (p = 0.9 using Chi-square test, see *Table 3*).

## Spatiotemporal distribution of *S. Typhi* cases and carriers

We examined the spatial and temporal distribution of all *S.* Typhi isolates collected at the study clinics (see methods; *Figure 3—figure supplement 1* and *Supplementary file 9*), and the subset of 96 *S.* Typhi from cases and 67 from carriers living within the demographic surveillance site (DSS) (*Figure 3*, and *Table 4*). A number of peaks in monthly *S.* Typhi case and carrier numbers are apparent in both cohorts (*Figure 3*, *Figure 3—figure supplement 1*), with fewer cases and carriers observed in warmer months. Carrier counts remained relatively consistent throughout the study period. We tested for association between case or carrier peaks ( > 2 positives per month) and high rainfall or temperature in the same month, previous month, or 2 months prior to the month of observation (*Table 4*, and *Supplementary file 9*). For those *S.* Typhi from within the DSS, high temperatures were associated with lower case and carrier counts in the same month, and in the subsequent month (p < 0.05, using Fisher's exact test), however no associations between high rainfall and elevated case or carrier counts was observed. Repeating these analyses with WGS-confirmed isolates, the association with temperature was no longer significant (although the trend remained; see *Supplementary file 10*, *Supplementary file 11*). No association was detected between patient numbers attending study clinics and climate variables (*Supplementary file 12*).

GPS coordinates were available for n = 139 (55%) *S.* Typhi isolates, and we endeavoured to

**Table 3.** Typhi genotypes associated with n = 153 cases and controls among different age groups.

| | Age group | | |
|---|---|---|---|
| | ≤ 1 year | 1–7 years | > 7 years |
| WGS-confirmed cases | 7 | 66 | 26 |
| EA1 | 5 (71%) | 24 (36%) | 6 (23%) |
| EA2 | 1 (14%) | 34 (52%) | 11 (42%) |
| EA3 | 0 | 8 (12%) | 7 (27%) |
| non-H58 | 1 (14%) | 0 | 2 (78%) |
| Shannon diversity | 0.80 | 0.97 | 1.25 |
| WGS-confirmed carriers | 4 | 30 | 20 |
| EA1 | 1 (25%) | 10 (33%) | 9 (45%) |
| EA2 | 3 (75%) | 16 (53%) | 8 (40%) |
| EA3 | 0 | 1 (3%) | 1 (5%) |
| non-H58 | 0 | 3 (10%) | 2 (10%) |
| Shannon diversity | 0.56 | 1.05 | 1.11 |

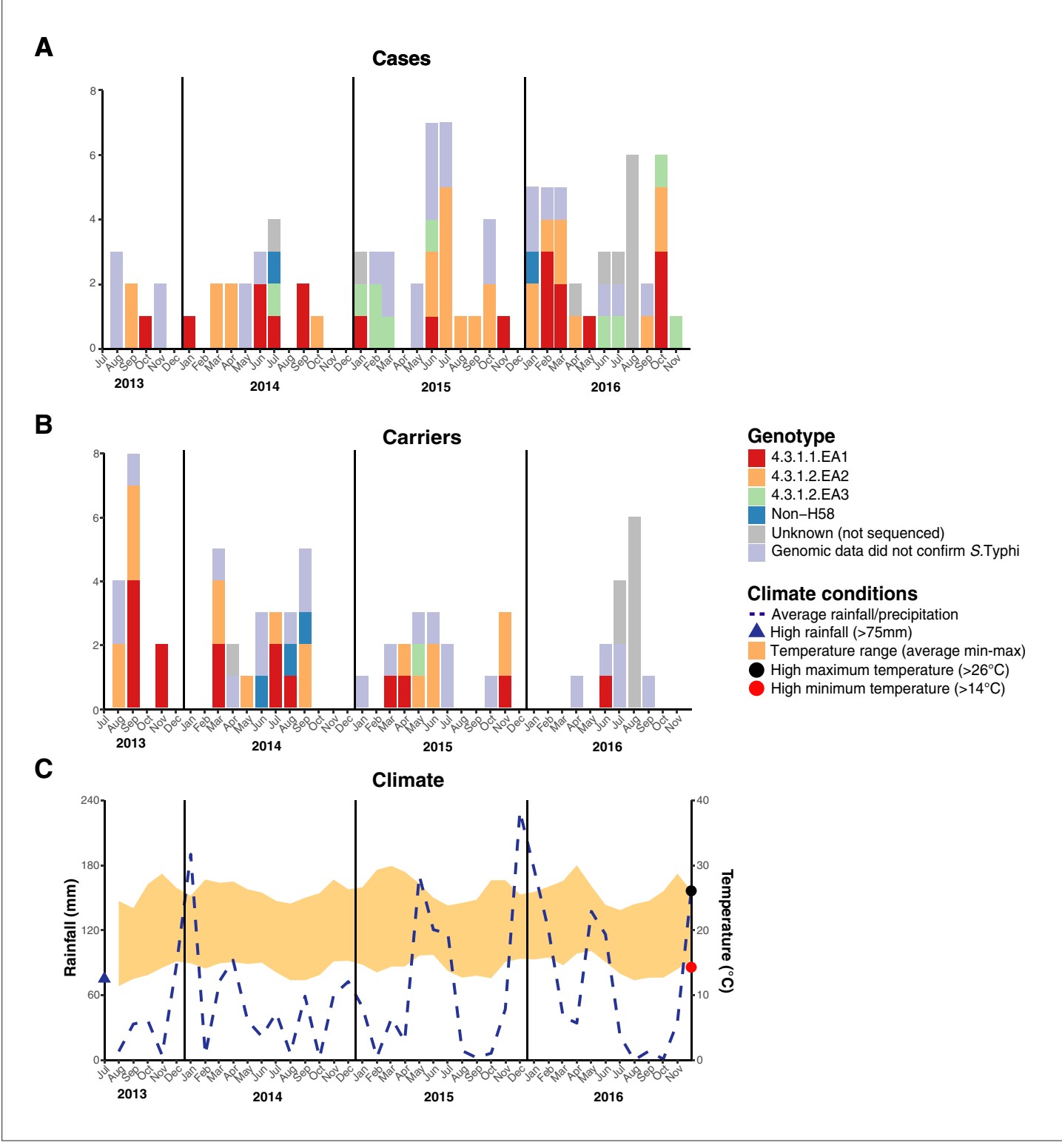

**Figure 3.** Epidemic curve of all *S. Typhi* cases and controls per month inside the DSS. (**A**) Monthly distribution of *S. Typhi* genotypes from cases. (**B**) Monthly distribution of *S. Typhi* genotypes from carriers. Note that the counts include all participants who were culture-positive for *S. Typhi* and also those who were culture-positive for other *Salmonella* but identified later by WGS as *S. Typhi*. (**C**) Weather conditions throughout the study period. Blue dashed line indicates precipitation level per month (rainfall), shaded orange polygon indicates the temperature range, red circle indicates threshold for high minimum temperature for statistical testing, black circle indicates threshold for high maximum temperature for statistical testing, blue triangle indicates threshold for high rainfall for statistical testing.

*Figure 3 continued on next page*

*Figure 3 continued*

The online version of this article includes the following figure supplement(s) for figure 3:

**Figure supplement 1.** Epidemic curve of all *S.Typhi* cases and controls per month.

look for geographic hotspots suggestive of major point-source single-genotype outbreaks in the informal settlement. However, our data revealed that the three H58 genotypes and non-H58 genotypes were co-circulating throughout the study area, with no evidence of geographic restriction of specific genotypes (see *Figure 4—figure supplement 1*). Further, we did not observe any spatially linked phylogenetic clusters of closely related sequences.

## Evolutionary history of *S. Typhi* cases and controls

We applied Bayesian phylodynamic analysis to all available Kenyan H58 genomes to estimate the dates of emergence of each of the East African lineages. The data showed temporal structure (see methods and *Figure 4—figure supplement 2*), and we estimated a genome-wide substitution rate of 0.8 SNPs per genome per year (95 % HPD, 0.1–1.0). This translates to a rate of $1.9 \times 10^{-7}$ genome-wide substitutions per site per year (95 % HPD = $1.5 \times 10^{-7}$-$2.2 \times 10^{-7}$). The novel EA1 isolates from this study (accounting for 35 % of cases and 37 % of controls) were intermingled with those sequenced previously from Kenya and were genetically diverse (median pairwise distance ~16 SNPs, interquartile range 12–27). This is indicative of a well-established EA1 *S*. Typhi population in Nairobi for which we estimate the mrca existed circa 1990 (95% HPD, 1981–1999) (see *Figure 4a*, *Figure 4—figure supplement 3*). The most common lineage was EA2 (48%), which also showed extensive diversity and we estimate emerged circa 1988–1990 (95 % HPD, 1978–1997) (see *Figure 4a*, *Figure 4—figure supplement 2*), earlier than the first recorded H58 Lineage II isolation in Kenya in 2004 *Kariuki et al., 2010*. We estimate the MDR fluoroquinolone non-susceptible lineage EA3, which accounts for just 11 % of isolates, arrived much more recently (Kenyan mrca circa 2012, 95% HPD 2009–2014) (see *Figure 4a*, *Figure 4—figure supplement 3*). The topology of the global H58 tree (*Figure 2*) supports South Asia as the most likely origin for EA3, with EA3 strains spreading between Kenya and Uganda, probably through the shared transport systems.

**Table 4.** Climatic predictors of elevated case and control counts inside the DSS.
Values in cells are odds ratios and p-values for Fisher's exact test between high case or control count ( > 2 per month) and high rainfall/temperature. * highlights p-values < 0.05.

**Typhoid** cases

| Month | Same month | | Previous month | | 2 months prior | |
|---|---|---|---|---|---|---|
| | OR (95% CI) | p-value | OR (95% CI) | p-value | OR (95% CI) | p-value |
| Rainfall (precipitation) > 75 mm | 0.21 (0.019–1.2) | 0.079 | 1.4 (0.26–6.9) | 0.73 | 3.7 (0.73–22.3) | 0.08 |
| Minimum temperature > 14 °C | 0.21 (0.041–0.95) | 0.025* | 0.61 (0.14–2.6) | 0.52 | 2.2 (0.49–10.5) | 0.33 |
| Maximum temperature > 26 °C | 0.85 (0.20–3.6) | 1 | 0.37 (0.080–1.6) | 0.20 | 0.67 (0.15–2.80) | 0.75 |

Asymptomatic controls

| Month | Same month | | Previous month | | 2 months prior | |
|---|---|---|---|---|---|---|
| | OR (95% CI) | p-value | OR (95% CI) | p-value | OR (95% CI) | p-value |
| Rainfall (precipitation) > 75 mm | 1.2 (0.21–6.5) | 1 | 0.43 (0.038–2.7) | 0.45 | 0.43 (0.038–2.7) | 0.45 |
| Minimum temperature > 14 °C | 0.12 (0.016–0.64) | 0.005* | 0.41 (0.078–1.9) | 0.30 | 0.65 (0.13–3.2) | 0.73 |
| Maximum temperature > 26 °C | 0.10 (0.0090–0.61) | 0.005* | 0.19 (0.027–1.0) | 0.04* | 0.63 (0.12–3.0) | 0.73 |

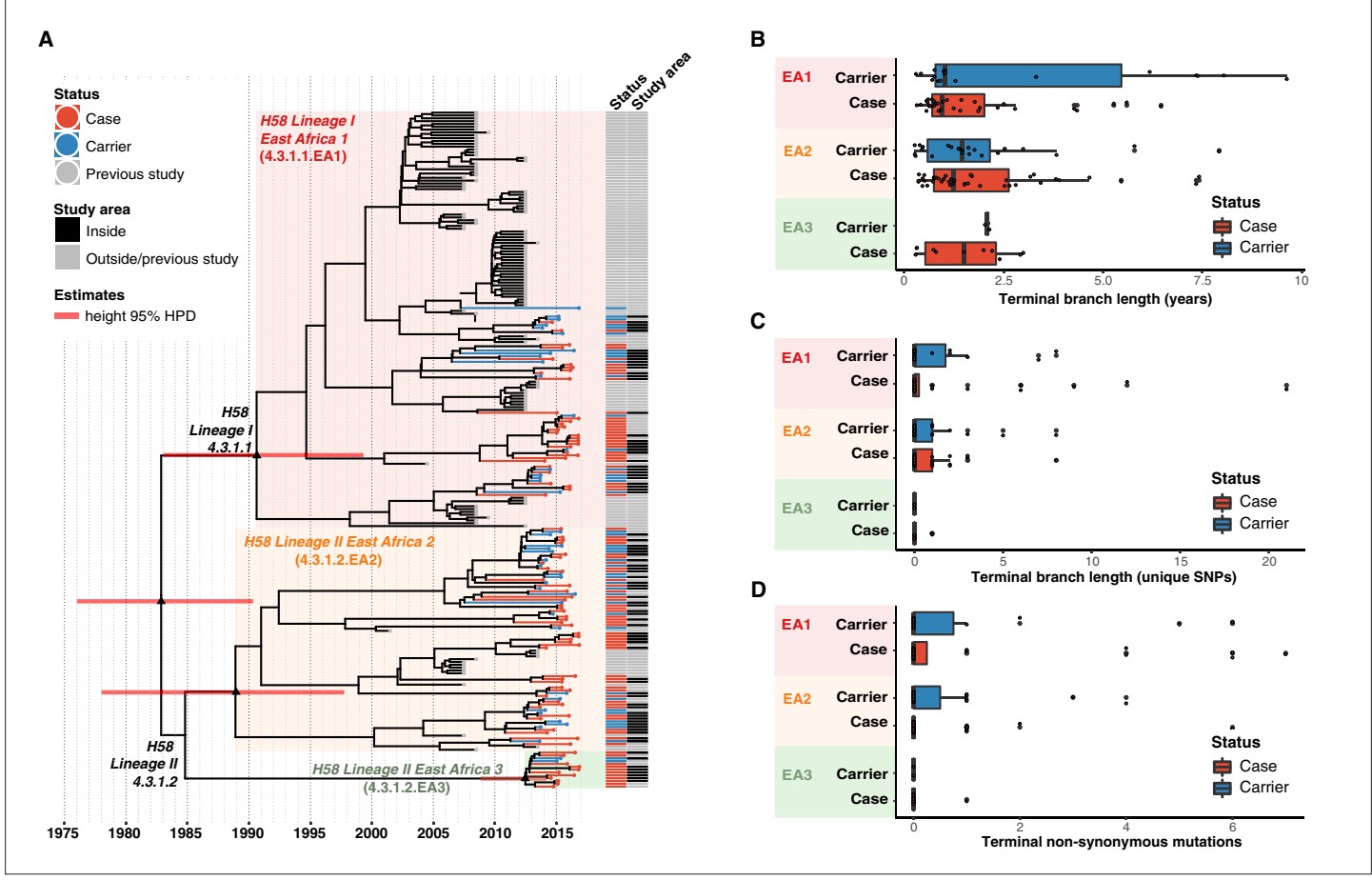

**Figure 4.** Temporal distribution of genotypes and among all cases and carriers. (**A**) Dated maximum-clade credibility phylogenetic tree of Kenyan *S.* Typhi genotype 4.3.1 (H58), including 128 isolated from this study. Tip colours & first colour bar indicate symptom status, second colour bar indicates those isolates from children living in the defined survey area. Black triangles demarcate nodes of interest, and the accompanying bars indicate 95% HPD of node heights. Interactive phylogeny available at https://microreactorg/project/I2KUoasUB. (**B**) Distribution of terminal branch lengths for all sequences, extracted from the Bayesian tree shown in (**A**). (**C**) Distribution of isolate-specific SNPs detected in sequences from all cases and controls. (**D**) Distribution of terminal non-synonymous mutations detected in sequences from all cases and controls. In the boxplots in panels B, C, and D, black bars indicate median values, boxes indicate interquartile range. Cases and carrier samples indicated as per the inset legend.

The online version of this article includes the following figure supplement(s) for figure 4:

**Figure supplement 1.** Spatial distribution of *S. Typhi* genotypes throughout the informal settlement.

**Figure supplement 2.** Tempest regressions & BEAST date randomisation testing.

**Figure supplement 3.** Temporal and age distribution of genotypes among cases and controls inside the survey site.

The Bayesian tree of Kenyan H58 isolates (***Figure 4a***, ***Figure 4—figure supplement 3***) shows intermingling of sequences from acute cases and asymptomatic carriers. Sequences from carriers appeared more deeply branched than those of cases which we tested by comparing the terminal branch lengths (estimated in units of time in the Bayesian phylogeny) and isolate-specific SNP counts, for high-quality H58 sequences from acute cases (n = 85) vs those of asymptomatic carriers (n = 43) (***Figure 4b***, ). The mean values were higher for carriers vs cases (***Figure 4b–c***, ***Figure 4—figure supplement 3***), with the trend being more pronounced among samples from within the DSS but these trends were not statistically significant (p = 0.42 for unique SNPs and p = 0.57 for terminal branches for all samples, using one-sided Wilcoxon rank sum test; p = 0.051 for unique SNPs and p = 0.58 for terminal branches in DSS). The mean number of non-synonymous (NS) mutations detected in terminal branches was greater for carrier isolates than those from cases, but again this difference was not statistically significant (0.72 vs 0.54 for all sequences, p = 0.53 using Wilcox rank sum test; 0.81 vs 0.39, p = 0.20 inside the DSS; see ***Figure 4d***, ***Figure 4—figure supplement 3***). There was also no

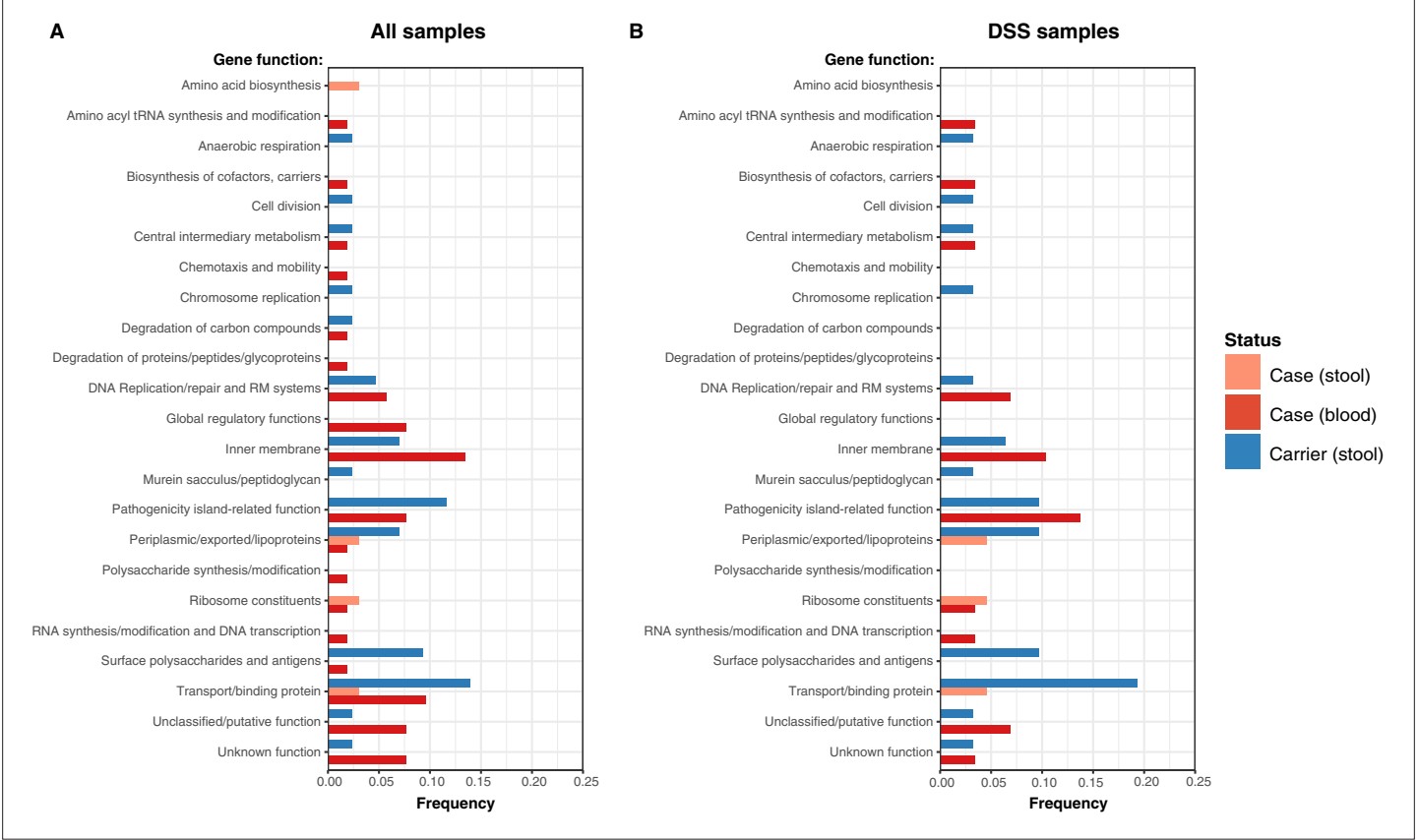

**Figure 5.** Frequency of terminal non-synonymous mutations in difference gene functional categories among cases and carriers. (**A**) Frequency of terminal non-synonymous mutations in all sequences collected. (**B**) Frequency of terminal non-synonymous mutations in sequences from within the DSS area. Red bars indicate the frequency non-synonymous mutations found in acute case samples from blood, peach bars indicate non-synonymous mutations found in acute case samples from stool, and blue bars indicate the frequency of mutations found in carrier samples from stool.

significant difference in terminal branch lengths or unique SNP counts between genomes carrying vs lacking MDR genes or QRDR mutations (data not shown).

Examination of the location of terminal-branch NS mutations revealed that certain functional categories of genes carried more NS mutations arising on terminal branches associated with carriage samples vs those from acute cases (*Figure 5*; *Supplementary file 13*). Notably, carriage samples were associated with significantly higher frequencies of terminal-branch NS mutations in genes responsible for the synthesis of surface polysaccharides and antigens (9.3 % of carriers vs 1.2 % of acute cases (all of which were blood isolates), p = 0.043, Fisher's exact test). Notably, in the *viaB* operon (responsible for Vi capsule biosynthesis) we identified n = 2/43 carriage isolates that harboured NS mutations (*tviD*-R159C and *tviE*-P263S) compared with only n = 1/85 case isolate (*tviB*-V1M); and in the *wba* cluster genes (responsible for O-antigen biosynthesis) we identified n = 2/43 carriage isolates that harboured NS mutations (*wza*-V137G and *wzxC*-L26F) whilst none were detected among case samples (*Supplementary file 13*). Non-significant excesses of mutations in carriage isolates were also observed for periplasmic and exported lipoproteins (6.98% vs 1.92 % in blood isolates from febrile cases and 3.03 % in stool isolates from febrile cases, see *Figure 5*).

## Discussion

In this case-control typhoid surveillance study, we observed an asymptomatic *S. Typhi* carriage rate of 1.1 % among children aged 16 years and under from an informal settlement with endemic Water, Sanitation, and Hygiene (WaSH)-related enteric diseases *Mbae et al., 2020*; *Kariuki et al., 2020*; *Kariuki et al., 2019*. To our knowledge, there has not been systematic surveillance for typhoid carriage in communities in Africa, but globally carriage and shedding of *S. Typhi* has mostly been associated

with older age groups *Levine et al., 1982*; *Senthilkumar et al., 2012*; *Gunn et al., 2014*. Our data highlights a role for paediatric carriage, revealing a lower percentage of carriers amongst infants ≤ 1 year of age (0.62%), increasing to 1.2 % in children between 7 and 16 years (*Supplementary file 6*). Thus, carriage and shedding, especially among school age children, is likely an important factor in the onward transmission of typhoid in this setting *Mbae et al., 2020*. Symptomatic typhoid fever is common in school age children, with a case culture positive rate of 4.3 % among febrile children 7–16 years of age, though our data shows that there is also a substantial burden among younger children 1–7 years of age, and infants up to 1 year of age, with culture-positive rates among febrile participants of 3.1% and 2.2%, respectively (*Supplementary file 6*). Our WGS data shows that serotyping frequently leads to mis-classification of *S*. Typhi vs non-typhoidal *Salmonella*, highlighting the importance of molecular diagnostics in future studies and surveillance programs. WGS has been widely adopted by *Salmonella* reference labs in high-income countries *Ashton et al., 2016*; *Köser et al., 2012*; *Arnold et al., 2018* but remains rare in low-income countries, although KEMRI now sequences all *Salmonella* isolates on-site immediately after serotyping.

We previously noted a dominance of MDR H58 *S*. Typhi over the last decade, essentially replacing the antimicrobial susceptible genotypes that dominated in the 1980–1990 s *Kariuki et al., 2015*. The *S*. Typhi circulating in the informal settlement in the present study are largely comprised of descendants of the previously observed H58 sublineages 4.3.1.EA1 (36%) and 4.3.1.2.EA2 (48%) *Wong et al., 2015*; *Park et al., 2018*. Both EA1 and EA2 appear to be long established genotypes, with the mrca of EA1 existing circa 1990 (95% HPD, 1981–1999) (see *Figure 4a* ), consistent with the earliest recorded detection of H58 Lineage I in Kenya in 1988 *Kariuki et al., 2010*. Similarly we predict that the mrca of EA2 existed circa 1988–1990 (95 % HPD, 1978–1997), earlier than the first recorded H58 Lineage II isolation in Kenya in 2004 *Kariuki et al., 2010*. Our data thus support contemporaneous imports of EA1 and EA2 in the late 1980 s or early 1990 s, shortly after the emergence of H58 in South Asia *Wong et al., 2015* (circa 1982, 95 % HPD 1974–1990), and show that both lineages have persisted and diversified locally alongside one another in the intervening decades. H58 sublineage EA3 was introduced later (we estimate that the Kenyan mrca existed circa ~2012 [95% HPD 2009 to 2014]), and consistent with this the lineage displays less diversity and accounts for a smaller fraction of cases and controls (11%). The topology of our global phylogeny (*Figure 2*) suggests that South Asia is the most likely origin of EA3 (as it is for EA1 and EA2), and that EA3 appears to have spread between Uganda and Kenya. This in line with multiple reports *Feasey et al., 2015*; *Park et al., 2018* of H58 strains spreading through East Africa, mainly arising from intracontinental and transcontinental travel and concomitant risk factors associated with WASH conditions.

The sublineages of *S*. Typhi H58 in Kenya exhibit different antibiotic resistance profiles (*Figure 2* and *Table 2*). Notably, EA1 has a large Kenyan sublineage of MDR strains with IncHI1 plasmids *Wong et al., 2015*; *Park et al., 2018* (which are commonly associated with outbreaks in East Africa and Asia *Wong et al., 2015*; *Park et al., 2018*; *Holt et al., 2011*) but also a Kenyan sublineage with chromosomally integrated MDR. Chromosomal integration of MDR has not previously been reported in *S*. Typhi from Kenya (see supplementary data), but has been reported in Malawi and Tanzania *Wong et al., 2015* and our new data suggests that the variant may have been transferred to these locations from Kenya (see *Figure 2*). MDR H58 isolates are now widespread across East Africa, having been detected in Malawi, Uganda, Rwanda, Tanzania, and Mozambique *Wong et al., 2015*; *Feasey et al., 2015*; *Park et al., 2018*; *Wong et al., 2016*; *Ingle et al., 2019*.

The three East African lineages differed markedly in their patterns of mutations conferring Decreased Ciprofloxacin Susceptibility (DCS). GyrB-S464F was conserved among all EA2, whereas all EA3 isolates carried the GyrA-S83Y mutation. The GyrA-S464F mutation was also detected at low frequency in EA1 (*Table 2*). This data indicates that ciprofloxacin resistance has been selected independently multiple times and is ongoing. Increasing rates of ciprofloxacin resistance have also been observed following similar introductions of H58 elsewhere in East Africa *Park et al., 2018*, and likely reflect a change in treatment practise following widespread dissemination of MDR *S*. Typhi elsewhere including South and Southeast Asia *Britto et al., 2018*; *Britto et al., 2020*; *Rahman et al., 2020*.

The different *S*. Typhi lineages appeared to be fairly evenly distributed between both acute cases and carriers, with the most common subgroup (EA2) accounting for 46 % of acute cases and 50 % of carriers. Similarly, all *S*. Typhi genotypes were identified throughout the study period and spatially across the study site, with most case/carrier monthly counts and geographic regions containing a

diversity of genotypes (*Figure 3*, *Figure 3—figure supplement 1*). Our data therefore provides no evidence for major point-source single-genotype outbreaks, but is consistent with persistent contamination of water supplies with multiple *S. Typhi* genotypes. Higher temperatures were associated with lower *S. Typhi* case and carrier counts, however, no association with high rainfall was observed among our culture data. These findings are in line with previous studies focusing on seasonal trends in nearby Kibera *Breiman et al., 2012*. However, they contrast with trends previously observed in other settings including Malawi, where higher temperatures and rainfall were associated with increased risk of disease albeit with a time lag of multiple months *Thindwa et al., 2019*, and South Asia *Baker et al., 2011*; *Dewan et al., 2013*; *Kanungo et al., 2008*.

In our phylogenetic trees, branch lengths and SNP counts can be interpreted as measures of evolutionary time, and thus terminal branches and isolate-specific SNP counts as a measure of time since that isolate shared a mrca with another sampled isolate. This total time includes (i) time from mrca to the acquisition of infection in the sampled host, plus (ii) time from initial acquisition of the infection to time of sampling (ie time within the sampled host). Variation in the latter is more likely to be explained by the symptom status of the sampled host (rather than the former which occurs prior to the infection of the sampled host). Hence the higher mean branch lengths and SNP counts in asymptomatic controls is consistent with the expectation that carriers have had, on average, a longer duration from acquisition of the infection to sampling in the clinic (which is triggered by routine visits and unrelated to symptoms or colonisation status), compared to acute cases who are presenting due to febrile illness triggered by a recently acquired S. Typhi infection. This supports the interpretation that *S. Typhi*-positive controls identified in this study represent genuine medium- to long-term typhoid carriers, rather than simply reflecting transient presence in the gut. The greater diversity observed here amongst controls (*Table 2*) further supports this interpretation. Longer branch lengths among carrier samples were also observed in a recent study *Thanh Duy et al., 2020* of *S. Typhi* isolated from bile samples from the gallbladders of cholecystectomy patients in Nepal. The differences in terminal branch lengths were non-significant in our study, possibly reflecting low statistical power or, perhaps less likely, that our control data constitute a mix of multiple carriage types including convalescent (3 weeks to 3 months), temporary (3e to 12 months), and chronic (more than 1 year) carriers. Also in line with previous findings *Thanh Duy et al., 2020*, our analyses provide evidence of positive selection among carriage isolates, with a higher proportion of non-synonymous mutations detected among carriers in specific biological pathways including surface polysaccharides and antigens, transport/binding proteins, and anaerobic respiration (*Figure 5*, *Supplementary file 13*). This is exemplified in the genes encoding surface antigens, notably those responsible for biosynthesis of Vi capsule and O-antigen lipopolysaccharide.

Our study is not without limitations, firstly, our data are from a single informal settlement community in Nairobi, and thus may not be representative of the overall population structure and AMR patterns of typhoid in Kenya more broadly, or in older age groups. Similarly, our sample size yielded a relatively small number of isolates for WGS, and we thus lack statistical power for some genetic analyses.

## Conclusion

Our study is the first case-control study to identify and sequence both typhoid carriers and cases contemporaneously in an endemic community setting. High rates of AMR among both infection types in Kenya combined with high carriage and case rates, especially in the younger age groups, highlight the need for enhanced AMR and genomic surveillance in this region to inform both treatment guidelines and control strategies that keep pace with the local evolution and spread of AMR. Intervention strategies are urgently needed including the introduction of the new Vi conjugate vaccine in a programme that includes targeting of paediatric age groups in the short term, and improvements to WaSH infrastructure in the long term.

## Acknowledgements

Research reported in this publication was supported by the National Institute of Allergy and Infectious Diseases of the National Institutes of Health under Award Number R01AI099525 to SK and the Wellcome Trust. ZAD was supported by a grant funded by the Wellcome Trust (STRATAA; 106158/Z/14/Z and TyVac), and received funding from the European Union's Horizon 2020 research and innovation

programme under the Marie Skłodowska-Curie grant agreement TyphiNET No 845,681. GD was supported by the Cambridge Biomedical Research Council NIHR AMR theme. KEH was supported by a Senior Medical Research Fellowship from the Viertel Foundation of Australia, and the Bill and Melinda Gates Foundation, Seattle (grant #OPP1175797).

## Additional information

### Funding

| Funder | Grant reference number | Author |
| --- | --- | --- |
| National Institutes of Health | R01AI099525 | Samuel Kariuki |
| Wellcome Trust | 106158/Z/14/Z | Gordon Dougan<br>Kathryn E Holt |
| European Commission | TyphiNET No 845681 | Zoe A Dyson<br>Kathryn E Holt |
| National Institute for Health Research | AMR Theme | Gordon Dougan |
| Bill and Melinda Gates Foundation | OPP1175797 | Kathryn E Holt |

The funders had no role in study design, data collection and interpretation, or the decision to submit the work for publication.

### Author contributions

Samuel Kariuki, Conceptualization, Formal analysis, Funding acquisition, Investigation, Project administration, Supervision, Validation, Writing – original draft, Writing – review and editing; Zoe A Dyson, Writing – original draft, Writing – review and editing; Cecilia Mbae, Investigation, Methodology, Project administration, Writing – review and editing; Ronald Ngetich, Susan M Kavai, Stephen Anyona, Naomi Gitau, Investigation, Methodology; Celestine Wairimu, Formal analysis, Methodology; Robert Sanaya Onsare, Formal analysis, Project administration; Beatrice Ongandi, Sebastian Duchene, Formal analysis; Mohamed Ali, Data curation, Formal analysis; John David Clemens, Conceptualization, Formal analysis, Writing – original draft, Writing – review and editing; Kathryn E Holt, Validation, Writing – original draft, Writing – review and editing; Gordon Dougan, Conceptualization, Methodology, Validation, Writing – original draft, Writing – review and editing

### Author ORCIDs

Samuel Kariuki http://orcid.org/0000-0003-3209-9503
Zoe A Dyson http://orcid.org/0000-0002-8887-3492

### Ethics

Human subjects: The study was approved by the Scientific and Ethics Review Unit (SERU) of the Kenya Medical Research Institute (KEMRI) (Scientific Steering Committee No. 2076). All parents and/or guardians of participating children were informed of the study objectives and voluntary written consent was sought and obtained before inclusion.

### Decision letter and Author response

Decision letter https://doi.org/10.7554/eLife.67852.sa1
Author response https://doi.org/10.7554/eLife.67852.sa2

## Additional files

### Supplementary files

• Supplementary file 1. *S. Typhi* sequences used in this study (excel file).

• Supplementary file 2. Outgroup and global context H58 *S. Typhi* sequences used in this study (excel file).

- Supplementary file 3. Kenyan sequences used in temporal analyses (excel file).
- Supplementary file 4. Sequences used in IncHI1 plasmid analyses (excel file).
- Supplementary file 5. Repetitive regions excluded from plasmid pAKU1 SNP analysis (excel file).
- Supplementary file 6. Culture positive typhoid cases and asymptomatic carriers.
- Supplementary file 7. Comparison of phenotypic and genotypic AMR profiles of 136 (n=128 H58, n=8 Non-H58) high quality *S. Typhi* genome sequences.
- Supplementary file 8. Distribution of n=153 *S. Typhi* genotypes among each sex for cases and controls.
- Supplementary file 9. Climatic predictors of elevated case and carrier counts for all samples.
- Supplementary file 10. Climatic predictors of WGS confirmed elevated case and carrier counts inside the DSS.
- Supplementary file 11. Climatic predictors of WGS confirmed elevated case and carrier counts for all samples collected.
- Supplementary file 12. Climatic predictors of elevated cases presenting at study clinics.
- Supplementary file 13. Non-synonymous (NS) Mutations among *S. Typhi* isolates (excel file).
- Transparent reporting form

## Data availability

All data generated or analysed during this study are included in the manuscript and supporting files. Raw Illumina sequence reads have been submitted to the European Nucleotide Archive (ENA) under accession PRJEB19289. Individual sequence accession numbers are listed in Table S1.

The following dataset was generated:

| Author(s) | Year | Dataset title | Dataset URL | Database and Identifier |
|---|---|---|---|---|
| Kariuki S, Dyson Z | 2020 | Typhi_Kenya European Nucleotide Archive (ENA) | https://www.ncbi.nlm.nih.gov/bioproject/?term=PRJEB19289 | NCBI BioProject, PRJEB19289 |

The following previously published datasets were used:

| Author(s) | Year | Dataset title | Dataset URL | Database and Identifier |
|---|---|---|---|---|
| Parkhill J, Dougan G, James KD | 2001 | Complete genome sequence of a multiple drug resistant Salmonella enterica serovar Typhi CT18 | https://www.nature.com/articles/35101607 | Nature, 35101607 |
| Argimon S, Abudahab K, Goater RJE | 2016 | Visualizing and sharing data for genomic epidemiology and phylogeography | https://www.ncbi.nlm.nih.gov/pmc/articles/PMC5320705/ | Microb Genom, PMC5320705 |

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

# Appendix 1

## Supplementary methods

### Phylogenetic and SNP analysis of *S.* Typhi isolates

For SNP analysis, paired-end reads from 153 *S.* Typhi isolates were mapped to the reference sequence of *S.* Typhi CT18 (accession number: AL513382) *Parkhill et al., 2001* using the RedDog mapping pipeline (v1beta.10.3), available at http://githibcom/katholt/reddog. Briefly, RedDog uses Bowtie (v2.2.3) *Langmead and Salzberg, 2012* to map reads to the reference sequence; SAMtools (v0.1.19) *Li et al., 2009* to identify SNPs with phred quality scores above 30; filter out SNPs supported by <5 reads, or with >2.5 times the genome-wide average read depth (representing putative repeated sequences), or with ambiguous (heterozygous) consensus base calls. For each SNP position that passed these criteria in any one isolate, consensus base calls (i.e. alleles) for that position were extracted from all genomes, and used to construct an alignment of alleles across all SNP sites. Ambiguous base calls and those with phred quality <20 were treated as unknown alleles and represented with a gap character in the SNP alignment. Read alignments were used to assign isolates to previously defined lineages according to the *S.* Typhi extended genotyping framework *Wong et al., 2016*; *Britto et al., 2018*; *Rahman et al., 2020*, by subjecting the alignments (BAM format) to analysis with the GenoTyphi pipeline (available at http://github.com/katholt/genotyphi). Unique SNPs defining three novel lineages were identified from the genome-wide SNP allele table (with SNPs responsible for non-synonymous mutations in highly conserved genes without deletions prioritized for lineage definitions), these were added to the GenoTyphi scheme to facilitate easy identification of these lineages in future studies.

Phylogenetic analyses were restricted to WGS-confirmed pure cultures of *S.* Typhi H58 (genotype 4.3.1, n = 128). For some analyses, an additional 1,076 *S.* Typhi H58 genomes from previously published WGS studies of global and African isolates *Wong et al., 2015*; *Park et al., 2018*; *Wong et al., 2016*; *Pham Thanh et al., 2016* were also included for context (using the same mapping approach detailed above). Alleles from 61 additional *S.* Typhi genomes representing all non-H58 subclades (listed in *Supplementary file 2*), and *S.* Paratyphi A str. AKU_12601 (accession FM200053) *Holt et al., 2009*, were also included in the phylogenetic analysis as outgroups for tree rooting. SNPs called in phage regions or repetitive sequences (354 kbp; ~ 7.4 %) of bases in the CT18 reference chromosome, as defined previously *Wong et al., 2015*; *Holt et al., 2008* were filtered from the alignment, which was then used with the CT18 reference genome (AL513382) to produce a whole genome pseudoalignment that was subjected to analysis with Gubbins (v2.3.2) *Croucher et al., 2015* to remove any further recombinant regions. This resulted in a final set of 8,635 SNPs identified from an alignment of 4,809,037 sites for the 1,266 isolates.

From the global SNP alignment, maximum likelihood (ML) phylogenetic trees were inferred using RAxML (v8.2.9) *Stamatakis, 2014*, with a generalized time-reversible model, a Gamma distribution to model site-specific rate variation (the GTR+ $\Gamma$ substitution model; GTRGAMMA in RAxML), and 100 bootstrap pseudo-replicates to assess branch support.

From the global alignment we extracted a separate SNP alignment for the set of 239 Kenyan *S.* Typhi H58 genomes (n = 128 from this study and n = 111) from published studies, see *Supplementary file 3*; *Park et al., 2018*; *Wong et al., 2016*, which had length 489 SNPs. The same phylogenetic inference methods were used to generate a ML tree from the SNP alignment of 239 Kenyan isolates for temporal analyses (described below).

Phylogenetic trees were visualized using Microreact (interactive global H58 phylogeny available at: https://microreact.org/project/wViqmaRdZuFVEb6yk4i1jU) *Argimon et al., 2016* and the R package *ggtree* v1.14.6 *Yu et al., 2016*. For the purpose of plotting the global H58 tree, clusters of *S.* Typhi isolates that were members of the same monophyletic clade and isolated from the same location in South East Asian countries (i.e. representing local outbreaks) were reduced to a single representative each using the *drop.tip*() function in the R package *ape* (v5.4.1) *Paradis et al., 2004*.

Terminal branch lengths were extracted from phylogenies using R package *ggtree Yu et al., 2016*. Pairwise SNP distances were calculated from alignments using the *dist.dna*() function in the R package *ape Paradis et al., 2004*. Non-synonymous mutations were detected using SNPPar (V0.4.2dev) *Edwards et al., 2020* and grouped by function based on the gene in which they were found according to the *S.* Typhi functional classification scheme developed at the Sanger Institute

(http://www.sanger.ac.uk/) using the genome annotation of CT18 *Thanh Duy et al., 2020*; *Parkhill et al., 2001*.

## SNP analysis of *S.* Typhi IncHI1 plasmids

In order to investigate the potential for plasmid transmission among the three East African *S.* Typhi H58 sublineages we carried out an SNP analysis of IncHI1 plasmid minor variants. Firstly, raw read data for all *S.* Typhi sequences included in the chromosomal SNP analysis described above were mapped to the reference sequence of IncHI1 plasmid pAKU_1 (accession number AM412236) using RedDog (as described above). Those plasmid sequences where a read depth of at least 10-fold and coverage across the refence sequence of at least 75 % were observed (n = 669 of sequences analysed; see *Supplementary file 4*) were included in SNP analysis, with repetitive regions excluded as detailed in *Supplementary file 5*. Plasmid MLST sequence types (pMLST) were determined using SRST2 *Inouye et al., 2014*; *Jolley and Maiden, 2010*; *Holt et al., 2011*; *Phan et al., 2009*, and we then interrogated the RedDog output allele matrix, and found that among n = 534 sequences of plasmid sequence type 6 (PST6), associated with genotype 4.3.1, there were a total of 25 variable sites. We concatenated these to form alignments, selected a single representative sequence of each plasmid haplotype, omitting those with gap characters, and calculated a genetic distance matrix using the R function *dist.dna*() from the R package *ape* *Paradis et al., 2004*. The distance matrix was then used as input for a Minimum Spanning Tree which was inferred using the R function *mst*() also in the R package ape, and visualised using the R package *ggnetwork* (v0.5.9) *Briatte, 2021*.

## Phylodynamic analysis of *S.* Typhi isolates

To investigate temporal signal and date the introduction of *S.* Typhi H58 into Kenya based on the 239 available genomes (n = 128 from this study, and n = 111 from previous studies *Park et al., 2018*; *Wong et al., 2016*), we used several methods. First, we used TempEst (v1.5.1) *Rambaut et al., 2016* to assess temporal structure (i.e. clock-like evolution) by conducting a regression analysis of the root-to-tip branch distances of the ML tree as a function of sampling date (expressed as decimal years at a resolution of days), using the heuristic residual mean squared method with the best fitting root selected. To estimate divergence dates for the three *S.* Typhi H58 sublineages we detected in Kenya (EA1-3), we used BEAST (v1.10) *Suchard et al., 2018* to fit a phylodynamic model to the SNP alignment and isolation dates (as decimal years at a resolution of days). Note that as EA1-3 almost exclusively comprise Kenyan strains in the global H58 tree and the parent node for each of EA1-3 is the parent node for all Kenyan isolates of EA1-3 (see *Figure 2*, *Figure 2—figure supplement 1*), the divergence dates for the EA1-3 parent nodes in the tree of 239 Kenyan isolates is taken as the divergence date for each of EA1-3 generally, as well as the lower bound for the date of introduction of each of these sublineages into Kenya.

We ran separate models using constant-coalescent population size and Bayesian skyline tree priors, in combination with a strict clock model or a relaxed (uncorrelated log normal distribution) clock model, to identify the best fitting model for our data. For BEAST analyses the GTR+$\Gamma$ substitution model was selected, and sampling times (tip dates) were used to calibrate the molecular clock (for isolates collected in this study the precise day of isolation was used; for the previously published genomes, only the isolation year was known, so tip dates were assigned to the first of July for that year with an uncertainty of 0.5 years). For all tree prior and model combinations, a chain length of 100,000,000 steps with sampling every 5,000 steps was used *Duchene et al., 2016*. The relaxed (uncorrelated lognormal) clock model, that allows for evolutionary rate variations among branches of the tree and the constant-coalescent model were found to best fit our data. To assess the temporal signal of these Bayesian estimates, we conducted a date-randomisation test where sampling times were assigned randomly to the sequences, and the analysis re-run 20 times with the best fitting models (constant-coalescent demographic and uncorrelated lognormal clock) *Duchene et al., 2016*; *Firth et al., 2010*. The date-randomisation test revealed that these data displayed 'strong' temporal structure (meeting the criterion CR2 of *Duchene et al., 2016*). Our preliminary BEAST runs resulted in implausible tree topologies and dated the most recent common ancestor (mrca) of H58 *S.* Typhi in Kenya ~ 1927 (95% highest posterior density = 1847–1984), conflicting with previously inferred divergence dates for the emergence of H58 (5,7,26). Fixing the tree topology to that obtained from ML inference yielded more plausible date estimates (~1968, 95 % HPD = 1957–1977), however, sampling from the prior (without the sequence alignment) using the same model showed that this estimate was driven

entirely by the priors provided to the model, with no information contributed by the sequence data. Taken together, these preliminary analyses suggested that while a temporal signal is present in the alignment, the signal is weak and would benefit from the specification of sensible priors to calibrate the root height. Previous analyses of the H58 divergence date, inferred using global data spanning a wider sampling period and with stronger temporal signal, estimate it emerged circa 1989 (95% HPD, 1981–1995) *Wong et al., 2015*, and we have previously reported the presence of H58 lineage one in Kenya in the late 1980s *Kariuki et al., 2010*. We therefore specified a log-normally distributed root height prior with mean 1989 and standard deviation 4 years. Use of this root height prior (without fixing the tree topology) yielded a plausible tree topology (i.e. consistent with the outgroup-rooted maximum likelihood tree inferred from the same alignment, with the expected separation of H58 sublineages into monophyletic clades), showed evidence of temporal signal (via date randomisation testing), and was not driven by priors alone; hence this approach was used for the final analyses presented here.

For the final analyses, 2 independent runs each conducted with a chain length of 100,000,000 steps sampling every 5,000 iterations were combined using LogCombiner (v1.10.0) *Suchard et al., 2018*, after removing the first 10 % of steps from each as 'burn-in'. Maximum-clade credibility trees (MCC) trees were generated with 'common ancestor heights' specified for node heights, using TreeAnnotator (v1.10.0) *Suchard et al., 2018*. The effective sample sizes (ESS) from the combined runs were >200 for all reported parameters. The resultant MCC tree was visualized using *ggtree* (v1.14.6) *Yu et al., 2016* and Microreact *Argimon et al., 2016* (interactive phylogeny available at: https://microreact.org/project/I2KUoasUB).

## Genomic determinants of antimicrobial resistance

The read mapping-based allele typer SRST2 (v0.2.0) *Inouye et al., 2014* was used to detect the presence of plasmid replicons (PlasmidFinder database *Carattoli et al., 2014*) and antimicrobial resistance (AMR genes) (ARGannot database *Gupta et al., 2014*) and to identify the precise alleles of AMR genes. Where AMR genes were observed without evidence of a known AMR plasmid, raw read data was de novo assembled using Unicycler (v0.4.7) *Wick et al., 2017* and then examined visually using the Bandage (v0.8.1) *Wick et al., 2015* assembly graph visualizer, in order to interrogate the assembly to confirm the chromosomal location and composition of AMR-associated transposons. As the Tn*2670*-like composite transposon commonly associated with the acquisition of MDR genes in *S.* Typhi is mediated by IS*1* translocation *Wong et al., 2015*, ISMapper (v2.0) *Hawkey et al., 2015* was used with default parameters to screen all read sets for insertion sites of transposases of IS*1* (accession number J01730) relative to the CT18 reference chromosome sequence, in order to identify the location of any such insertion sequences in the chromosome of each Kenyan *S.* Typhi genome. Single point mutations located within the quinolone resistance determining region (QRDR) of genes *gyrA*, *gyrB*, and *parC* associated with reduced susceptibility to fluoroquinolones *Pham Thanh et al., 2016* were detected from the whole genome read alignments (BAM files) described above, using GenoTyphi *Wong et al., 2016*; *Britto et al., 2018*; *Rahman et al., 2020*. Where a resistance phenotype was detected in the absence of known molecular determinants of AMR, DBGWAS (v0.5.4) *Jaillard et al., 2018* was utilised to carry out a bacterial genome-wide association study (GWAS) to screen for genetic loci and/or variants associated with the observed resistance phenotype.

## Statistical and spatial analysis

All statistical analyses unless otherwise stated were carried out using R (v4.0.2). Multivariate logistic regression analyses were conducted with the *glm*() function in base R, and linear regression analyses of were carried out using the *lm*() function in base R. Shannon diversity was calculated using the *diversity*() function in the R package *vegan Dixon, 2003*. Base R function *chisq.test*() was used to conduct a Chi-Squared test of age groups and *S.* Typhi genotypes. Fisher's exact test was carried out using the *fisher.test*() function in base R to investigate the frequency of non-synonymous mutations among cases and carriers, as well as associations between binary categories ('high' vs 'low') for monthly case counts or carrier counts and weather conditions (monthly rainfall, minimum and maximum temperature) in either the same month, the previous month, or two months prior. Thresholds for the 'high' categories were: case or carrier counts >2; rainfall > 75 mm; minimum temperature >14 °C; maximum temperature >26 °C. For analyses of elevated patient counts attending study clinics, we used the same climate thresholds, but instead used an elevated patient count threshold of n = 85.

GPS coordinate data were visualized using Microreact *Argimon et al., 2016*.

## Supplementary results

### Precision of AMR phenotypes from genotypes

Observed AMR phenotypes for all high quality WGS data (n = 128 H58 and n = 8 Non-H58 genome sequences) were largely explained by the presence of known molecular determinants of AMR (*Supplementary file 7*). Sensitivity was generally high for previous first line drugs for treating typhoid including chloramphenicol, ampicillin, co-trimoxazole, as well as tetracyclines ( > 94%), as was specificity ( > 74%), and low very major errors (VME; failure to predict a resistance phenotype) were determined for these drugs ( < 4%). Major errors (ME; failure to predict a susceptible resistance phenotype) ranged between 5.1%–7.4% for these drugs, and for three sequences, this is potentially explained by the loss of an IncHI1 plasmid (n = 2) or Tn*2670*-like transposon (n = 1) associated with the MDR phenotype in culture, as resistance phenotypes were not observed for any of these drugs.

VME for nalidixic acid (32.4%), and ME (23.5%) for ciprofloxacin were both high. The sensitivity and specificity for these drugs (*Supplementary file 7*) are confounded by the presence of the GyrB-S464F mutation (*Table 2*), conserved in all n = 73 genome sequences of genotype 4.3.1.2.EA2 (H58 lineage two sublineage East Africa 2; discussed below in detail). The GyrB-S464F mutation has been previously reported to cause Decreased Ciprofloxacin Susceptibility (DCS), but not nalidixic acid resistance *Accou-Demartin et al., 2011*, however, in our data the relationship between GyrB-S464F Quinolone/Fluoroquinolone resistance in this strain background remains unclear as 50.7 % of EA2 isolates showed reduced susceptibility to nalidixic acid and 47.9 % ciprofloxacin, with the rest testing sensitive to these drugs. Neither comparative analysis of resistant and sensitive EA2 sequences, nor further bacterial GWAS analysis, revealed any evidence of either causal or compensatory mutations, or the acquisition/loss of genetic loci responsible for these phenotypes, perhaps due to low power.

Specificity estimates were high (100%), but sensitivity low (0%), for third generation cephalosporins including ceftazidime cefotaxime ceftriaxone and cefpodoxime (see *Supplementary file 7*). This was the result of phenotypic resistance to 3$^{rd}$ generation cephalosporins detected at low frequency among our high quality sequenced genomes (n = 1–4, 0.73–2.9%) in the absence of any known molecular determinants for resistance to these drugs. A GWAS was again carried out using susceptibility data for these drugs, but no candidate molecular mechanisms significantly associated with the observed resistance phenotypes were identified, again possibly due to low power.

### Local subpopulations of *S*. Typhi H58

*Salmonella* Typhi H58 (genotype 4.3.1) is subdivided into lineages I (genotype 4.3.1.1) and II (genotype 4.3.1.2). Lineage II was more common in this setting than lineage I: n = 90 (62.1% of H58) vs n = 55 (37.9%). Examination of the global phylogeny (*Figure 2* and online interactive version https://microreact.org/project/wViqmaRdZuFVEb6yk4i1jU) revealed all H58 lineage I isolates from this study shared a most recent common ancestor (mrca) whose descendants form a monophyletic clade that exclusively comprised *S*. Typhi from East African countries (see *Figure 2*). This clade corresponds to the previously reported introduction of H58 lineage I from South Asia into Eastern Africa, which appears to have arrived in Kenya before spreading to Tanzania and on to Malawi and South Africa (*Feasey et al., 2015*; *Wong et al., 2016*; *Figure 2*). Here, we define this as H58 sublineage EA1 (East Africa 1) with genotype designation 4.3.1.1.EA1 (labelled in *Figure 2*), which can be identified by the presence of a synonymous marker SNP STY0750-G1407A (position 751,854 in CT18; this genotype has been added to the GenoTyphi scheme available at http://github.com/katholt/genotyphi).

*S*. Typhi H58 lineage II (genotype 4.3.1.2) isolates from our study belonged to two distinct clades of the global phylogeny (*Figure 2*), which were each exclusively populated by East African isolates. The largest of these clades (n = 80 isolates, of which 81.3 % derive from the current study) formed a monophyletic group in which isolates from this study were intermingled with those obtained from previous studies in Kenya (n = 14, 17.5%), and a single Tanzanian strain isolated in 2012 (see *Figure 2*), suggestive of a single inter-country transmission event from Kenya into Tanzania. This clade is nested within a deeper clade of diverse South Asian isolates (see *Figure 2*), and corresponds to the previously reported introduction of H58 lineage II into Kenya from South

Asia *Wong et al., 2015*; *Park et al., 2018*. This lineage, here defined as H58 sublineage EA2 (East Africa 2) and designated genotype 4.3.1.2.EA2 (labelled in *Figure 2*), can be identified by the presence of a synonymous marker SNP STY4818-C1069T (position 4680610 in CT18). The smaller East African H58 lineage II clade (n = 43 isolates) comprised two sister clades, separated by ≥13 SNPs, one involving isolates from Kenya (n = 13, all from this study) and the other isolates from Uganda (n = 30, which accounted for 100 % of the typhoid burden at the Ugandan site in the TSAP study *Park et al., 2018*) (see *Figure 2*). This clade (including both the Ugandan and Kenyan subgroups) is here defined as H58 sublineage EA3 (East Africa 3) and designated genotype 4.3.1.2.EA3 (labelled in (2) *Figure 2*), identified by synonymous marker SNP STY2750-G96A (position 2587488 in CT18). Both EA2 and EA3 genotypes have been added to the GenoTyphi scheme.

The three East African H58 subgroups circulating in our setting all had high rates of MDR (84%, 74% and 94%, respectively); however in EA2, MDR was exclusively associated with the PST6-IncHI1 plasmid, and in EA3 exclusively with the chromosomal insertion (see *Table 2*). In EA1, most MDR was associated with the PST6-IncHI1 plasmid. However, a subclade of isolates (associated with spread to Tanzania and Malawi) carried the chromosomal insertion instead (see *Table 2*). Chromosomal integration of MDR strains has not previously been reported in *S.* Typhi from Kenya, although it has been reported in the Malawi and Tanzanian sublineages of EA1 *Wong et al., 2015* which our data suggests transferred to those locations from Kenya. Here, the integration of the MDR composite transposon into the chromosome of Kenyan EA1 and EA3 isolates was supported by both examination of genome assembly graphs and analysis of IS*1* insertion sites (see Methods), both of which supported integration near gene *cyaA* as has been reported previously *Wong et al., 2015*; *Britto et al., 2018*. Interestingly, MDR in the Ugandan subclade of EA3 was associated with the IncHI1-PST6 plasmid, suggesting that migration of the transposon from plasmid to chromosome may have occurred in situ in Kenya after divergence from the Ugandan branch. The three East African lineages also differed markedly in their patterns of mutations associated with reduced susceptibility to fluoroquinolones: GyrB-S464F was conserved among all EA2, whereas all EA3 isolates carried the GyrA-S83Y mutation, and three distinct GyrA mutations, and the GyrA-S464F mutation were detected at low frequency in EA1 (see *Table 2*).

