## [Decision Letter]

**Acceptance summary:**

This study contributes a significant advance to the field in terms of detailed genomic epidemiology of the introductions of drug resistant typhoid into Kenya, and Africa in general. The findings highlight the role of asymptomatic carriage in the spread, drug-resistance mechanisms and the emergence of typhoid strains in Kenya. The authors also contribute a small number of sequences from both carriage and acute disease, most sequenced cases are associated with acute disease, making this deposit of publicly available carriage sequences extremely valuable. This work will be of interest to a broad readership, including but not limited to clinicians, biologists, and public health experts.

**Decision letter after peer review:**

Thank you for submitting your article "Multiple introductions of multidrug-resistant typhoid associated with acute infection and asymptomatic carriage, Kenya" for consideration by *eLife*. Your article has been reviewed by 3 peer reviewers, and the evaluation has been overseen by a Reviewing Editor and Bavesh Kana as the Senior Editor. The following individuals involved in review of your submission have agreed to reveal their identity: Lauren Cowley (Reviewer #2); Madikay Senghore (Reviewer #3).

Essential revisions:

As noted by the reviewers, the authors present a novel and high quality study elucidating further aspects of the role of asymptomatic carriage within typhoid epidemiology and placing East African lineages in the geographic context of introductions from south Asia. Most recommendations from the reviewers are for clarifications; a modest list of suggested analyses for the revision follows before the public reviews and evaluation summary:

1) One major factor that is overlooked in the investigation, is the sampling of acute cases both from blood and fecal culture. In the reviewers' opinion, these should be considered as separate conditions and separate sample types (so not grouped together for some of the analysis.) This would affect results in figure 5 that could be further separated out into three groups; controls, enteric cases and invasive cases. It might be expected that positive selection had happened in the differing environments of the acute cases, either in invasive blood cultured specimens or enteric fecal cultured specimens that could be investigated and detailed (see further details on this point in the comments from Reviewer 2)

(2) Another aspect of the paper that could have borne more scrutiny was the PTS6-IncHI1 plasmid present in two of the east African lineages. Although short read sequencing limits its resolution or the level of comparative analysis possible, some attempt at comparing the short reads generated in this study to previously published full sequences of PTS6-IncHI1 plasmids in H58 Typhi would be warranted. This could inform horizontal transfer of this plasmid and whether these plasmids sampled in two separate lineages in the same location contemporaneously were highly similar or not.

(3) The authors rightly mention that recombinant regions were excluded in the genomes prior to phylogenetic tree construction. The reviewers wondered if the phylogenies with and without recombinant regions filtered out could be compared. This would provide a sense of the overall impact of recombination in the evolution of the S. Typhi strains. Also, it's not clear if the recombination events removed in the phylogeny of only Kenyan isolates, specific clades and global isolates. Considering that Gubbins was developed to identify recombination events in clonal bacterial population, the authors should comment on whether it is appropriate to run the analysis using a diverse global dataset. It would also be great to mention the recombination rate inferred from Gubbins.

*Reviewer #2 (Recommendations for the authors):*

I really enjoyed reading this paper and think that, on the whole, the analysis produced and conclusions drawn are very well justified and carefully prepared. I have just two small suggestions to provide further detail to this paper that I think would be extremely valuable to the readers.

(1) Separate the acute infections into two groups, those sampled from fecal culture and those from blood culture. This is an important distinction that would be important to the analysis performed to produce figure 5. There could be a distinct signal in NS mutations from the fecal vs. blood acute samples. Furthermore, the carriage fecal and acute fecal samples could have mutations that differentiate them from the blood culture samples or vice versa. Some consideration of this or inclusion of this in your analysis would be worthwhile.

(2) To interrogate the PTS6-IncHI1 plasmid further, I suggest blasting the contigs from your plasmid-identified samples against previously published and finished PTS6-IncHI1 plasmid sequences. You might be able to piece together more about the content of this plasmid and how similar the PTS6-IncHI plasmids from EA1 are to those from EA2. Otherwise some similar style of analysis that investigates horizontal transfer further.

*Reviewer #3 (Recommendations for the authors):*

The authors present a well written manuscript based on a case contact study of diarrheal aetiology among children in Kenya. They implement robust bioinformatics pipelines and derive conclusions that can guide intervention strategies. Below are some specific comments.

Line 159 – The authors mentions that for carriers, rectal swabs or stools samples were cultured on selenite F, however, it is not specified whether the isolates from cases were recovered using the same procedures. Please clarify

Line 185-186. Could the authors explain why there is such a huge discrepancy and how they ensured that this was not the result of sample mix-ups.

Table 1 – the authors use the results from culture to determine whether there are correlates for S. typhi carriage and disease. Given the discrepancies between the genomic data and the serotyping data why did the authors choose to use the serotyping data instead of the genomic confirmation. Are the results of this analysis reproducible if the genome confirmed data are used instead of the serotyping data?

Line 406 – The authors present evidence suggesting that higher temperatures were significantly associated with decreased case counts. However, this includes isolates that were not confirmed by sequencing and generally the per month case counts were low. Was there a correlation between weather and the general numbers of patients presenting at the clinic? Additionally, when the unconfirmed isolates are excluded from the analysis does this observation still hold?

---

## [Author Response]

Essential revisions:As noted by the reviewers, the authors present a novel and high quality study elucidating further aspects of the role of asymptomatic carriage within typhoid epidemiology and placing East African lineages in the geographic context of introductions from south Asia. Most recommendations from the reviewers are for clarifications; a modest list of suggested analyses for the revision follows before the public reviews and evaluation summary:(1) One major factor that is overlooked in the investigation, is the sampling of acute cases both from blood and fecal culture. In the reviewers' opinion, these should be considered as separate conditions and separate sample types (so not grouped together for some of the analysis.) This would affect results in figure 5 that could be further separated out into three groups; controls, enteric cases and invasive cases. It might be expected that positive selection had happened in the differing environments of the acute cases, either in invasive blood cultured specimens or enteric fecal cultured specimens that could be investigated and detailed (see further details on this point in the comments from Reviewer 2)

As detailed in the Methods section, we define acute typhoid cases as children with ≥3 days fever ≥38°C and positive blood or stool culture for S. Typhi (see lines 137-141). We include stool culture within this definition because blood culture is known to be only ~50% sensitive (see e.g. Antillon et al. 2018, J Infect Dis, doi: 10.1093/infdis/jiy471), and the presence of febrile illness with culture-proven presence of S. Typhi is generally considered to represent an acute infection as opposed to asymptomatic carriage. However, we agree with the reviewer’s point that for the purposes of examining positive selection in the bacteria, there could potentially be a distinction between selective pressure on isolates recovered from stool vs blood of febrile cases. We have therefore reanalysed the data as suggested and updated Figure 5, which now shows the data stratified into three groups: (i) stool isolates from asymptomatic controls; (ii) blood isolates from febrile cases; and (iii) stool isolates from febrile cases. The results show that, for the functional groups noted in text (surface polysaccharides and lipoproteins), the stool isolates from cases are similar to blood isolates and distinct from stool isolates from asymptomatic carriers.

We have updated the Results section (lines 548-551) to reflect this:

“Notably, carriage samples were associated with significantly higher frequencies of terminal-branch NS mutations in genes responsible for the synthesis of surface polysaccharides and antigens (9.3% of carriers vs 1.2% of acute cases (all of which were blood isolates), p=0.043, Fisher’s exact test). […] Non-significant excesses of mutations in carriage isolates were also observed for periplasmic and exported lipoproteins (6.98% vs 1.92% in blood isolates from febrile cases and 3.03% in stool isolates from febrile cases, see Figure 5).”

(2) Another aspect of the paper that could have borne more scrutiny was the PTS6-IncHI1 plasmid present in two of the east African lineages. Although short read sequencing limits its resolution or the level of comparative analysis possible, some attempt at comparing the short reads generated in this study to previously published full sequences of PTS6-IncHI1 plasmids in H58 Typhi would be warranted. This could inform horizontal transfer of this plasmid and whether these plasmids sampled in two separate lineages in the same location contemporaneously were highly similar or not.

To address this, we carried out an additional SNP analysis of n=669 IncHI1 plasmid sequences detected in the first global typhoid genomics study (Wong et al. 2015, Nat Genet, doi: 10.1038/ng.3281) and a multi-site surveillance study in East and West African (Park et al., 2018, Nat Commun, doi: 10.1038/s41467-018-07370-z) also included in the Typhi SNP analyses (sequences are listed in the new Supplementary Files 2 and 4). Results are shown in Figure 2 and new Figure 2—figure supplement 1.

We have added an additional section to the Results section titled ‘Global population structure and antimicrobial resistance profiles of Kenyan S. Typhi’, from lines 362-366 that now reads:

“We compared single nucleotide variant haplotypes for these plasmids with those from 534 IncHI1 PST6 plasmids sequenced previously from African and global studies (all of which were carried by H58 S. Typhi hosts, see Supplementary File 4). […] This is consistent with ongoing microevolution of the PST6 plasmid within S. Typhi lineages since the acquisition of the plasmid by the mrca of S. Typhi H58, but shows no evidence of transfer of plasmid haplotypes between S. Typhi lineages.”

The Methods section (lines 282-286) was updated as follows:

“SRST2 was used to determine IncHI1 plasmid MLST (multi-locus sequence types) sequence types (pMLST), and minor alleles were identified by mapping to the plasmid pAKU1 reference sequence (accession number AM412236) in the same manner as described above for the S. Typhi chromosome (details in supplementary methods and Supplementary Files 4-5).”

Supplementary methods (new section ‘SNP analysis of S. Typhi IncHI1 plasmids’ lines 1173-1190) provides more details of the analysis:

“SNP analysis of S. Typhi IncHI1 plasmids

In order to investigate the potential for plasmid transmission among the three East African S. Typhi H58 sublineages we carried out an SNP analysis of IncHI1 plasmid minor variants. […] The distance matrix was then used as input for a Minimum Spanning Tree which was inferred using the R function mst() also in the R package ape, and visualised using the R package ggnetwork (v0.5.9) ^23^.”

(3) The authors rightly mention that recombinant regions were excluded in the genomes prior to phylogenetic tree construction. The reviewers wondered if the phylogenies with and without recombinant regions filtered out could be compared. This would provide a sense of the overall impact of recombination in the evolution of the S. Typhi strains. Also, it's not clear if the recombination events removed in the phylogeny of only Kenyan isolates, specific clades and global isolates. Considering that Gubbins was developed to identify recombination events in clonal bacterial population, the authors should comment on whether it is appropriate to run the analysis using a diverse global dataset. It would also be great to mention the recombination rate inferred from Gubbins.

*Salmonella* Typhi is a genetically monomorphic pathogen (a single lineage within *S. enterica*, comprising a single clonal complex in the MLST scheme) and has very low recombination rates (Holt et al. 2008 Nat Genet, doi: 10.1038/ng.195, 40(8); Achtman 2008 Annual Review of Microbiology, 62, doi: 10.1146/annurev.micro.62.081307.162832). Further, the H58 (4.3.1) genotype is considered a single clone (Wong et al. 2015, Nat Genet, doi: 10.1038/ng.3281; Roumagnac et al. 2006 Science, doi: 10.1126/science.1134933) and thus suitable for recombination filtering using Gubbins. Gubbins analysis of H58 (Kenya or global isolates) consistently identifies two regions of recombination occurring at the root of the H58/genotype 4.3.1 branch (see Phandango screenshot in Author response image 1). The first of these spans CT18 (accession AL513382) reference sequence coordinates 954,115–970,731 (genes STY0961-STY0976), and the second spans 1,438,676-1,467,273 (genes STY1485-STY1508). Including these two recombinant patches in the SNP alignment results in lengthening the relevant branch but has no meaningful impact on the inferred relationships between lineages or isolates. Therefore, all phylogenies reported in the manuscript are inferred from the alignment that excludes these gubbins identified regions.

**Author response image 1. sa2fig1:** 

We have updated the Methods section to clarify this (lines 234-245):“Analysis with Gubbins (v2.3.2) identified two regions affected by recombination (coordinates 954,115–970,731 (genes STY0961-STY0976), and 1,438,676-1,467,273 (genes STY1485-STY1508) in the CT18 reference genome), which were excluded from the alignment^40^.”

Reviewer #2 (Recommendations for the authors):I really enjoyed reading this paper and think that, on the whole, the analysis produced and conclusions drawn are very well justified and carefully prepared. I have just two small suggestions to provide further detail to this paper that I think would be extremely valuable to the readers.(1) Separate the acute infections into two groups, those sampled from fecal culture and those from blood culture. This is an important distinction that would be important to the analysis performed to produce figure 5. There could be a distinct signal in NS mutations from the fecal vs. blood acute samples. Furthermore, the carriage fecal and acute fecal samples could have mutations that differentiate them from the blood culture samples or vice versa. Some consideration of this or inclusion of this in your analysis would be worthwhile.

This comment has been addressed as detailed above (please see ‘Essential revisions’).

(2) To interrogate the PTS6-IncHI1 plasmid further, I suggest blasting the contigs from your plasmid-identified samples against previously published and finished PTS6-IncHI1 plasmid sequences. You might be able to piece together more about the content of this plasmid and how similar the PTS6-IncHI plasmids from EA1 are to those from EA2. Otherwise some similar style of analysis that investigates horizontal transfer further.

This comment has been addressed as detailed above (please see ‘Essential revisions’).

Reviewer #3 (Recommendations for the authors):The authors present a well written manuscript based on a case contact study of diarrheal aetiology among children in Kenya. They implement robust bioinformatics pipelines and derive conclusions that can guide intervention strategies. Below are some specific comments.Line 159 – The authors mentions that for carriers, rectal swabs or stools samples were cultured on selenite F, however, it is not specified whether the isolates from cases were recovered using the same procedures. Please clarify.

All isolates, whether from cases or carriers were recovered using the same procedure of pre-enrichment and selection in Selenite-F. We have modified lines 163-177 to now read:

“For blood culture, 1-3 mL for children <5 years of age and 5-10 mL for those 5-16 years of age was collected in a syringe, placed into Bactec media bottles (Becton-Dickinson, New Jersey, USA), incubated at 37^o^C in a computerized BACTEC 9050 Blood Culture System (Becton-Dickinson), and subcultured after 24-48 h onto blood, chocolate and MacConkey agar (Oxoid, Basingstoke, UK) plates. All isolates, whether from cases or carriers were cultured on selenite F (Oxoid) broth aerobically at 37^o^C overnight.”

Line 185-186. Could the authors explain why there is such a huge discrepancy and how they ensured that this was not the result of sample mix-ups.

This was a 3-year study and isolates were archived immediately after biochemical and serological ID to await WGS processing later, hence we believe it is more likely that we had mixed colonies in the initial culture. Regardless, we report our results both in terms of sero-identified, and WGS-confirmed, S. Typhi and there is little difference except in terms of sample size and thus statistical power for testing (see Tables 1 and Figure 4—Figure Supplement 3).

Table 1 – the authors use the results from culture to determine whether there are correlates for S. typhi carriage and disease. Given the discrepancies between the genomic data and the serotyping data why did the authors choose to use the serotyping data instead of the genomic confirmation. Are the results of this analysis reproducible if the genome confirmed data are used instead of the serotyping data?

We prioritised culture data for the logistic regression analyses (Table 1) because (a) this is the largest sample available, as the WGS process introduces a large amount of data loss not because of mis-classification but because many isolates were unable to be revived or failed sequencing (see Figure 1); and (b) it represents the primary classification that is available in clinical microbiology laboratories in most settings and therefore comparable to other studies, in contrast to WGS-confirmed. We did also show results for linear regression of infection rates on age, stratified by sex, for WGS-confirmed as well as serologically-defined infections, in Table S4 (now Supplementary File 6) which show the same associations.

However to improve clarity and robustness, we have now added logistic regression analyses for WGS-confirmed S. Typhi to Table 1, which shows the same associations with age amongst males. We have also modified the text at lines 323-326 to note this:

“Significant associations identified from culture data were also observed when using WGS confirmed infections only, and no significant statistical association was found between phenotypic or genotypic AMR patterns and case/control status, age, or sex.”

Line 406 – The authors present evidence suggesting that higher temperatures were significantly associated with decreased case counts. However, this includes isolates that were not confirmed by sequencing and generally the per month case counts were low. Was there a correlation between weather and the general numbers of patients presenting at the clinic? Additionally, when the unconfirmed isolates are excluded from the analysis does this observation still hold?

The monthly count of febrile cases presenting to the clinics was not associated with rainfall or temperature, this data is now shown in Supplementary File 12.

The choice to use culture confirmed data for the climate analysis was made for the same reasons as outlined above for the logistic regression analyses. We have repeated the analyses using WGS confirmed isolates and added two new tables to the revised manuscript (Supplementary Files 11-12). We have adjusted the text in lines 468-472 to refer to these results, as follows:

“Repeating these analyses with WGS-confirmed isolates, the association with temperature was no longer significant (although the trend remained; see Supplementary Files 10-11). No association was detected between patient numbers attending study clinics and climate variables (Supplementary File 12).”

And have added the below lines 1303-1305 to the Supplementary Methods section:

“For analyses of elevated patient counts attending study clinics, we used the same climate thresholds, but instead used an elevated patient count threshold of n=85.”